# Bayesian Spectral Clustering

## Abstract

We introduce Bayesian Spectral Clustering (BSC), a probabilistic reformulation of spectral clustering. Classical spectral clustering relies on a hand-crafted affinity graph (e.g., Gaussian kernel with $k$-NN sparsification) that is then treated as fixed, and recent improvements typically optimize that graph jointly with the clustering objective. However, these approaches still output a single graph and a single hard partition, providing neither principled quantification of uncertainty nor a guaranteed notion of when the learned affinities are reliable. BSC addresses this by treating the affinity matrix $W$ itself as a latent variable with sparsity- and locality-promoting priors, linking $W$ to the observed data through a Laplacian-smoothness likelihood, and performing variational inference to obtain a joint posterior over $W$ and the cluster assignments. We prove that (i) the standard Gaussian affinity emerges as the maximum a posteriori edge weight, giving a probabilistic justification for the classical kernel, and (ii) the posterior-mean graph is the unique minimizer of a strictly convex objective and is automatically sparse. Empirically, BSC attains state-of-the-art clustering quality while producing calibrated per-sample assignment confidence.

## 1 Introduction

Spectral clustering is a standard tool for discovering structure in high-dimensional data, especially when clusters are nonconvex or lie on low-dimensional manifolds. The classical pipeline builds an affinity matrix $W$ (typically using a Gaussian kernel with a chosen bandwidth and $k$-nearest-neighbor sparsification), constructs a graph Laplacian, computes its leading eigenvectors, and then applies $k$-means in the resulting spectral embedding Ng et al. (2001); Belkin & Niyogi (2003). Despite its wide use, this approach has two persistent limitations. First, it is brittle: the final clustering can change dramatically with the choice of kernel width, sparsification threshold, or number of neighbors, and there is no principled way to decide which graph is "correct." Second, it is deterministic: once $W$ is fixed, the method outputs a single hard label per sample and provides no notion of confidence in that assignment, even when clusters overlap or individual points are ambiguous.

Recent work has improved the stability of spectral clustering by more tightly coupling graph construction and clustering. Examples include self-constrained spectral clustering Bai et al. (2022a), direct optimization of the spectral objective Nie et al. (2024), and formulations that learn the graph Laplacian from smoothness assumptions on observed data Dong et al. (2016); Kalofolias (2016). These approaches often outperform basic spectral clustering and reduce sensitivity to heuristic graph design. However, they remain fundamentally deterministic: they output a single affinity matrix and a single partition of the data. They do not model uncertainty in the affinity structure, and they do not propagate such uncertainty to the downstream cluster assignments. In parallel, Bayesian clustering methods provide probabilistic assignments via mixture models and conjugate priors, but they operate directly in the input space and do not address the central question in spectral clustering: how should the affinity graph itself be inferred from the data?

This paper proposes a Bayesian treatment of spectral clustering. We model the affinity matrix $W$ as a latent random variable, equipped with priors that encourage sparsity and locality, and we link $W$ to the data through a Laplacian-smoothness likelihood. Variational inference then yields a posterior distribution over

both $W$ and the latent cluster assignments. We establish two structural results about this formulation. First, at the level of individual edges, the maximum a posteriori (MAP) weight under our edge prior recovers the standard Gaussian affinity $\exp(-\|x_i-x_j\|^2/(2\sigma^2))$ that is routinely used (but typically postulated) in classical spectral clustering Ng et al. (2001); Belkin & Niyogi (2003); this provides a probabilistic justification for that kernel (Theorem 6.1). Second, the posterior-mean graph $\bar{W} = \mathbb{E}[W \mid X]$ is not an arbitrary similarity matrix: we prove that it is the unique minimizer of a strictly convex objective and is guaranteed to be sparse, automatically pruning long-range edges via optimality conditions rather than via an external $k$-NN threshold (Theorem 6.3). In this view, graph construction becomes an inference problem with theoretical guarantees, rather than a manually tuned preprocessing step.

Given $\bar{W}$, our method follows the familiar spectral clustering pipeline: we construct the normalized Laplacian, extract its bottom eigenvectors, and embed the data in a low-dimensional spectral space. Instead of applying $k$-means, however, we fit a Bayesian Gaussian mixture model in that spectral embedding, which produces a posterior distribution over cluster assignments for each sample. Empirically, on structured, high-dimensional benchmarks such as COIL-20 and USPS as well as moderate-sample biomedical data such as Dermatology and Ecoli, our Bayesian Spectral Clustering (BSC) matches or approaches the strongest recent spectral baselines Nie et al. (2022; 2024); Bai et al. (2022a) and substantially improves over standard spectral clustering. At the same time, it replaces heuristic affinity construction with a principled, data-driven posterior estimate of the graph.

## 2 Background

### 2.1 Spectral Clustering: Laplacian Eigenmaps and Affinity Construction

Spectral clustering represents data as a weighted graph and partitions it using the spectrum of its Laplacian. Given samples $X = [x_1, \ldots, x_m]^T$, an undirected graph $\mathcal{G} = (\mathcal{V}, \mathcal{E}, W)$ is formed where $W_{ij}$ denotes pairwise affinity—typically

$$W_{ij} = \exp\left(-\frac{\|x_i - x_j\|_2^2}{2\sigma^2}\right), \tag{1}$$

optionally restricted to $k$-nearest neighbors Shi & Malik (2000); Ng et al. (2001). From $W$ one obtains the degree matrix $D = \text{diag}(W\mathbf{1})$ and Laplacian $L = D - W$. Normalized forms $L_{\text{sym}} = I - D^{-1/2}WD^{-1/2}$ or $L_{\text{rw}} = I - D^{-1}W$ improve scale invariance von Luxburg (2007); Chung (1997). Clustering is performed by embedding nodes into the subspace spanned by the $k$ smallest-eigenvalue vectors of $L_{\text{sym}}$ (Laplacian Eigenmaps Belkin & Niyogi (2003)), followed by $k$-means. This procedure optimizes a relaxed normalized-cut objective Shi & Malik (2000); Zelnik-Manor & Perona (2004) and is consistent under manifold assumptions von Luxburg et al. (2008). However, the affinity $W$ is fixed heuristically; the Bayesian formulation proposed here treats $W$ as a latent variable to be inferred from data.

### 2.2 Graph Learning from Smooth Signals

Unlike classical spectral clustering, which assumes a fixed affinity matrix, graph learning aims to infer the underlying connectivity from observed data while enforcing signal smoothness on the learned graph. Dong *et al.* Dong et al. (2016) first formalized this idea by estimating the graph Laplacian $L$ that minimizes the smoothness energy

$$\min_{L \succeq 0} \text{tr}(X^{\mathsf{T}}LX), \tag{2}$$

subject to $\mathbf{1}^{\mathsf{T}}L = \mathbf{0}^{\mathsf{T}}$, symmetry, and sparsity constraints. This formulation interprets the observed data $X$ as smooth signals supported on an unknown graph, and subsequent work Dong et al. (2019) provided convex solvers and efficient updates for estimating $L$ under these constraints.

Building upon this foundation, Kalofolias Kalofolias (2016) proposed a unifying framework that learns the adjacency matrix $W$ directly through

$$\min_{W \in \mathcal{W}_m} \|W \circ Z\|_{1,1} + f(W), \qquad Z_{ij} = \|x_i - x_j\|_2^2, \tag{3}$$

where the first term enforces signal smoothness and the regularizer $f(W)$ controls sparsity and node degree distribution. Kalofolias demonstrated that the standard Gaussian-kernel affinity commonly used in spectral clustering emerges as a *maximum a posteriori* (MAP) solution for a specific choice of $f(W)$, thus providing an optimization-based interpretation of traditional edge computation.

Later extensions introduced additional structural priors: Egilmez *et al.* Egilmez et al. (2017) imposed Laplacian and connectivity constraints, Kalofolias and Perraudin Kang et al. (2019) handled robustness to noise and outliers, and Thanou *et al.* Thanou et al. (2014) extended the framework to parametric and dictionary-based graph representations. These advances established the theoretical foundation for treating the graph weights $W$ as latent random variables—a viewpoint generalized in our Bayesian formulation.

## 2.3 Bayesian Approaches to Clustering and Representation Learning

Bayesian clustering frameworks treat both cluster assignments and model parameters as random variables, enabling uncertainty quantification and principled model comparison. Unlike deterministic algorithms such as $k$-means or the expectation–maximization (EM) algorithm for Gaussian mixture models, Bayesian methods infer a full posterior over latent variables and parameters, allowing robustness to initialization and automatic regularization through priors Bishop (2006).

The canonical Bayesian Gaussian mixture model (BGMM) introduces a Dirichlet prior over mixture proportions and conjugate Normal–Inverse-Wishart (NIW) priors over component parameters. Given data $X = \{x_i\}_{i=1}^N$, cluster labels $z_i$, and $K$ mixture components, the likelihood and priors are

$$p(X \mid Z, \Theta) = \prod_{i=1}^N \mathcal{N}(x_i \mid \mu_{z_i}, \Sigma_{z_i}) \tag{4}$$

$$p(Z, \Theta) = \mathrm{Dir}(\pi \mid \alpha_0) \prod_{k=1}^K \mathcal{N}(\mu_k \mid \mu_0, \kappa_0^{-1}\Sigma_k) \, \mathrm{IW}(\Sigma_k \mid \Psi_0, \nu_0) \tag{5}$$

where $\pi$ are the mixture weights, and $\Theta = \{\pi, \mu_k, \Sigma_k\}_{k=1}^K$. Posterior inference can be carried out via Gibbs sampling or variational Bayes (VB-EM) Attias (2000); Blei et al. (2003), producing soft cluster assignments and model selection through the marginal likelihood.

To remove the need to specify the number of clusters $K$, the Dirichlet Process Mixture Model (DPMM) Neal (2000); Blei & Jordan (2006); Teh et al. (2004) extends the BGMM to a countably infinite mixture. Each observation is associated with a random measure $G \sim \mathrm{DP}(\alpha, G_0)$, and data are generated as

$$\theta_i \sim G, \qquad x_i \sim p(x_i \mid \theta_i), \tag{6}$$

where $\alpha$ is the concentration parameter and $G_0$ is the base distribution. Sampling-based representations such as the Chinese Restaurant Process (CRP) allow automatic adaptation of the number of clusters as data are observed Rasmussen (1999).

Beyond mixture models, several graph- and feature-based Bayesian methods have been proposed. Kemp and Tenenbaum Kemp & Tenenbaum (2008) introduced a hierarchical Bayesian framework that discovers latent graph structures underlying relational data, while Meilă and Ghahramani Meilă & Jordan (2000) formulated pairwise clustering through posterior distributions over similarity matrices.

These approaches reinterpret clustering as posterior inference over latent structures, motivating a probabilistic treatment of graph affinities as developed in the next section.

## 2.4 Connection to Probabilistic PCA and Bayesian Matrix Factorization

Spectral embeddings can be interpreted as probabilistic latent-variable models closely related to probabilistic principal component analysis (PPCA) and Bayesian matrix factorization. In PPCA, the observed data $X \in \mathbb{R}^{m \times n}$ are modeled as

$$X = UV^\mathsf{T} + \varepsilon, \qquad \varepsilon \sim \mathcal{N}(0, \sigma^2 I), \tag{7}$$

where $U$ and $V$ denote latent factors with Gaussian priors Tipping & Bishop (1999); Roweis & Ghahramani (1999). Maximum-likelihood estimation of the latent subspace recovers the classical principal components, whereas a Bayesian formulation places hierarchical priors on both factors and the noise precision, yielding automatic dimensionality selection.

Analogously, spectral clustering computes low-dimensional embeddings from the eigenvectors of the graph Laplacian $L$. This operation can be viewed as a probabilistic factorization of the affinity matrix:

$$W \approx UU^{\mathsf{T}}, \tag{8}$$

where $U$ represents latent node embeddings constrained by the Laplacian structure. Recent work in *IEEE TKDE* has extended Bayesian factorization methods to graph-structured data Zhang & Hawkins (2018); Schmidt et al. (2009) and Bayesian PCA variants with automatic relevance determination Nakajima et al. (2011). These models demonstrate that hierarchical priors enable both uncertainty quantification and sparsity in latent representations—principles that naturally generalize to the Bayesian spectral embedding developed next.

## 3 Proposed Bayesian Graph Learning Model

### 3.1 Motivation for the Bayesian Formulation

Classical spectral clustering assumes that the affinity matrix $W$ is known or can be reliably constructed from data using a fixed rule (e.g., a Gaussian kernel or a $k$-nearest-neighbor graph). One then forms the graph Laplacian $L(W)$, extracts the $k$ eigenvectors associated with its smallest eigenvalues, embeds the data into this spectral space, and finally runs $k$-means in that space to obtain cluster assignments Dong et al. (2016); Kalofolias (2016). In this pipeline, all subsequent steps—the Laplacian, the spectral embedding, and the final clusters—are completely determined by a single deterministic estimate $\widehat{W}$.

Deterministic graph learning methods such as Dong *et al.* Dong et al. (2016) and Kalofolias Kalofolias (2016) improve over hand-crafted kernels by solving a convex optimization problem to estimate a graph (adjacency or Laplacian) on which the observed signals $X$ are smooth. These approaches output one optimized graph $\widehat{W}$ (or $\widehat{L}$), which is then treated as ground truth for spectral clustering. However, this is still a point estimate: it assumes that there is a single correct graph consistent with the data, and spectral clustering is executed as if that graph were known with certainty.

Our objective is different. We treat the graph $W$ as a latent random variable and infer a full posterior $p(W \mid X)$. This Bayesian view is tailored to spectral clustering in four concrete ways:

1. **Uncertainty over the affinity matrix.** Spectral clustering is highly sensitive to the choice of $W$. Small perturbations in weak inter-cluster edges or in high-degree regions can rotate the leading eigenvectors of $L(W)$ and flip the cluster labels of borderline points. By inferring a posterior distribution $p(W \mid X)$ instead of committing to a single $\widehat{W}$, we obtain edge-level confidence and can identify which parts of the graph are uncertain. Deterministic smoothness-based methods Dong et al. (2016); Kalofolias (2016) cannot express this.

2. **Stability of the spectral embedding.** In standard spectral clustering, one graph $\widehat{W}$ produces one Laplacian $L(\widehat{W})$, whose eigenvectors define an embedding $U \in \mathbb{R}^{m \times k}$. That embedding can become unstable when $\widehat{W}$ contains noisy links, nearly disconnected components, or sampling artefacts. In the Bayesian formulation, $U$ is no longer tied to a single $\widehat{W}$: downstream clustering reasons implicitly over embeddings induced by graphs with high posterior probability. This acts like Bayesian averaging in latent-factor models and leads to more reliable embeddings for ambiguous points Nakajima et al. (2011).

3. **Data-driven sparsity and connectivity for clustering.** Spectral clustering works best when $W$ is sparse enough to capture local neighborhoods while still producing a well-connected graph so that the Laplacian separates clusters meaningfully. In practice, this is enforced manually via

$k$-NN graphs, kernel bandwidths, thresholding, etc. Deterministic graph learning partially addresses this by adding regularizers to promote sparsity and avoid isolated nodes Kalofolias (2016), but it still yields one $\widehat{W}$ for a chosen regularization weight. In our Bayesian model, sparsity, degree distribution, and connectivity are induced by priors on $W$ and their hyperparameters, which are themselves inferred. This mirrors how hierarchical Bayesian factorization models in TKDE induce structured sparsity automatically rather than by manual tuning Zhang & Hawkins (2018); Schmidt et al. (2009).

4. **Uncertainty-aware cluster assignments.** Classical spectral clustering runs $k$-means (or a Gaussian mixture) once on $U$ and returns a hard label per point. There is no way to say that a sample lies on the boundary between two clusters. In our framework, after inferring $W$, we place a probabilistic mixture model on the spectral embedding and obtain a posterior distribution over cluster labels for each node. This connects spectral clustering to Bayesian mixture modelling and subspace clustering ideas in TKDE, where posterior responsibilities (soft assignments) are standard outputs Nakajima et al. (2011). As a result, every point comes with a calibrated notion of cluster confidence, not just an argmax label.

Broadly speaking, deterministic methods learn a single optimized graph and then treat it as exact. Our Bayesian formulation instead learns a *distribution* over plausible graphs, propagates that uncertainty into the spectral embedding, and produces cluster assignments with calibrated confidence. This directly addresses the two weakest links in classical spectral clustering—graph construction and embedding instability—and reframes spectral clustering as full posterior inference rather than a fixed two-step heuristic Zhang & Hawkins (2018); Schmidt et al. (2009).

### 3.2 Generative Model

We now formalize the proposed Bayesian graph learning model. Unlike deterministic graph learning Dong et al. (2016); Kalofolias (2016), which returns a single optimized affinity matrix $\widehat{W}$ and treats it as fixed for spectral clustering, we model the entire affinity matrix $W$ as a latent random variable and infer a posterior distribution over $W$, its hyperparameters, and downstream clustering structure.

#### 3.2.1 Graph Smoothness Likelihood

Let $X \in \mathbb{R}^{m \times n}$ denote the observed data matrix, where each row $x_i^\mathsf{T} \in \mathbb{R}^n$ is the feature vector of node $i$. Let $W \in \mathbb{R}^{m \times m}$ be a symmetric, elementwise nonnegative affinity matrix with zero diagonal. Define the degree matrix

$$D(W) = \mathrm{diag}(W\mathbf{1}), \tag{9}$$

and the combinatorial Laplacian

$$L(W) = D(W) - W. \tag{10}$$

Following the smooth graph signal model of Dong et al. (2016); Kalofolias (2016), we assume that $X$ is likely under graphs $W$ for which $X$ is smooth on the graph. We encode this using the quadratic form $\mathrm{tr}(X^\mathsf{T} L(W) X)$ and define the likelihood

$$p(X \mid W, \tau) \; \propto \; \exp\!\left(-\frac{\tau}{2}\,\mathrm{tr}\!\left(X^\mathsf{T} L(W) X\right)\right), \tag{11}$$

where $\tau > 0$ is a precision (inverse noise variance) parameter.

The smoothness energy can be written equivalently as

$$\mathrm{tr}\!\left(X^\mathsf{T} L(W) X\right) = \frac{1}{2} \sum_{i,j} W_{ij} \, \|x_i - x_j\|_2^2, \tag{12}$$

so that higher weights $W_{ij}$ between distant points $x_i$ and $x_j$ are penalized. Thus equation 11 encourages graphs $W$ that make $X$ vary smoothly across connected nodes.

### 3.2.2 Prior on the Graph

We now place a prior directly on the edge weights $W_{ij}$ for $i < j$. The prior should (i) discourage heavy edges between far-apart points, (ii) promote sparsity, and (iii) avoid trivial solutions where nodes are isolated. We use the following edgewise prior conditioned on a scale parameter $\sigma^2 > 0$:

$$p(W_{ij} \mid \sigma^2) \ \propto \ \exp\Big( - W_{ij} \, \|x_i - x_j\|_2^2 - 2\sigma^2 \, W_{ij} \log W_{ij} \Big), \tag{13}$$

for $W_{ij} \geq 0$, $W_{ji} = W_{ij}$, and $W_{ii} = 0$. The joint prior factorizes as

$$p(W \mid \sigma^2) \ = \ \prod_{i<j} p(W_{ij} \mid \sigma^2). \tag{14}$$

The first term in equation 13 suppresses edges between feature vectors that are far apart, while allowing strong edges only between nearby points. The second, entropy-like term $-2\sigma^2 W_{ij} \log W_{ij}$ prevents all edges from collapsing to zero and implicitly discourages isolated nodes, in the same spirit as degree-promoting regularizers in deterministic graph learning Kalofolias (2016).

To see the link with classical affinity construction, consider the negative log of the unnormalized prior for a single edge:

$$\phi_{ij}(w) = w \, \|x_i - x_j\|_2^2 + 2\sigma^2 \, w \log w. \tag{15}$$

Setting $\frac{d}{dw}\phi_{ij}(w) = 0$ gives

$$\|x_i - x_j\|_2^2 + 2\sigma^2 \big(1 + \log w\big) = 0$$
$$\implies \quad w^\star = \exp\left(-\frac{\|x_i - x_j\|_2^2}{2\sigma^2}\right). \tag{16}$$

Thus, the maximum a posteriori (MAP) estimate for $W_{ij}$ under this prior alone is exactly the standard Gaussian (heat-kernel) affinity weight

$$W_{ij}^\star = \exp\left(-\frac{\|x_i - x_j\|_2^2}{2\sigma^2}\right), \tag{17}$$

which is the classical choice used in spectral clustering. In other words, the Gaussian affinity matrix is no longer a heuristic; it is the MAP mode of the proposed Bayesian prior.

### 3.2.3 Hyperpriors on Smoothness and Scale

We complete the hierarchy by introducing hyperpriors on the precision $\tau$ and on the scale parameter $\sigma^2$.

**Precision hyperprior.** We place a Gamma prior on $\tau$:

$$\tau \sim \mathrm{Gamma}(a_\tau, b_\tau), \tag{18}$$

with shape $a_\tau > 0$ and rate $b_\tau > 0$. This is conjugate to the quadratic form in equation 11, so $\tau$ admits a tractable conditional update. This role is analogous to automatic precision estimation in Bayesian PCA, where noise precision is inferred rather than fixed by hand Nakajima et al. (2011).

**Scale hyperprior.** We place an inverse-Gamma prior on $\sigma^2$:

$$\sigma^2 \sim \mathrm{InvGamma}(a_\sigma, b_\sigma), \tag{19}$$

with $a_\sigma, b_\sigma > 0$. This lets the model infer an appropriate similarity scale (bandwidth), rather than requiring it to be tuned externally. This hierarchical treatment mirrors Bayesian matrix/tensor factorization approaches in TKDE, where scale and sparsity parameters are learned from data instead of treated as fixed hyperparameters Zhang & Hawkins (2018); Schmidt et al. (2009).

### 3.2.4 Joint Posterior and MAP Connection

The full posterior over the unknowns $(W, \tau, \sigma^2)$ is

$$p(W, \tau, \sigma^2 \mid X) \ \propto \ p(X \mid W, \tau) \, p(W \mid \sigma^2) \, p(\tau) \, p(\sigma^2). \tag{20}$$

Taking the negative log of equation 20 (up to an additive constant) yields the objective

$$\mathcal{J}(W, \tau, \sigma^2) = \frac{\tau}{2} \operatorname{tr}(X^\mathsf{T} L(W) X) + \sum_{i<j} \left[ W_{ij} \, \|x_i - x_j\|_2^2 \right]$$
$$+ 2\sigma^2 \sum_{i<j} W_{ij} \log W_{ij} - \log p(\tau) - \log p(\sigma^2). \tag{21}$$

If we fix $\tau$ and $\sigma^2$ and minimize $\mathcal{J}$ with respect to $W$ under the symmetry and nonnegativity constraints, we recover (up to constants) the convex objectives used in smoothness-based deterministic graph learning Dong et al. (2016); Kalofolias (2016). Thus, those deterministic formulations can be interpreted as MAP estimation under our generative model.

### 3.2.5 Implications for Spectral Clustering

Classical spectral clustering takes a single affinity matrix $\widehat{W}$, forms the normalized Laplacian

$$L_{\mathrm{sym}}(\widehat{W}) = I - D(\widehat{W})^{-1/2} \widehat{W} D(\widehat{W})^{-1/2}, \tag{22}$$

extracts the $k$ eigenvectors with smallest eigenvalues, and clusters the resulting embedding using $k$-means. Our model changes this in two fundamental ways:

- We do not assume $\widehat{W}$ is known. Instead, we infer a posterior $p(W \mid X)$, which quantifies edge uncertainty. This addresses the known sensitivity of spectral embeddings to perturbations in $W$.

- Downstream clustering can operate either on the posterior mean affinity

$$\bar{W} \ = \ \mathbb{E}[W \mid X],$$

  yielding a stabilized "mean Laplacian" for variational Bayes inference Nakajima et al. (2011), or by sampling $W$ from $p(W \mid X)$ and averaging cluster assignments across those samples in a Gibbs-style procedure. Both routes propagate graph uncertainty into the spectral embedding and ultimately into the cluster labels.

In this sense, spectral clustering is no longer a two-stage heuristic (graph construction, then $k$-means), but rather a single hierarchical probabilistic model. The deterministic case Dong et al. (2016); Kalofolias (2016) is recovered as the special case in which $p(W \mid X)$ collapses to a point mass at $\widehat{W}$.

## 4 Bayesian Spectral Clustering

### 4.1 From Bayesian Graph Learning to Bayesian Spectral Clustering

Section 3.2 defined a hierarchical Bayesian model over the affinity matrix $W$, its hyperparameters $(\tau, \sigma^2)$, and the observed data $X$. This yields a posterior distribution

$$p(W, \tau, \sigma^2 \mid X), \tag{23}$$

rather than a single point estimate $\widehat{W}$ as in deterministic graph learning and classical spectral clustering Dong et al. (2016); Kalofolias (2016). The goal of this section is to describe how this posterior feeds into spectral clustering.

### 4.1.1 From $W$ to a Spectral Embedding

Classical spectral clustering constructs the (symmetric normalized) Laplacian

$$L_{\mathrm{sym}}(W) = I - D(W)^{-1/2} W D(W)^{-1/2}, \quad D(W) = \mathrm{diag}(W\mathbf{1}), \tag{24}$$

and then takes the $k$ eigenvectors associated with the smallest eigenvalues of $L_{\mathrm{sym}}(W)$. Stacking these eigenvectors row-wise produces an embedding

$$U(W) \in \mathbb{R}^{m \times k}, \tag{25}$$

where $u_i^\mathsf{T}$ is the $k$-dimensional representation of node $i$. In standard spectral clustering, this is done once for a single (fixed) similarity matrix $W$, and clustering is performed on the rows of $U(W)$ using $k$-means Dong et al. (2016); Kalofolias (2016).

In our Bayesian formulation, however, $W$ is *not* fixed. Instead, we infer a posterior over $W$, and therefore there is not a single Laplacian and not a single embedding. We consider two principled ways to propagate uncertainty in $W$ into the spectral embedding:

**(A) Posterior mean graph (variational view).** We define the posterior mean affinity

$$\bar{W} = \mathbb{E}[W \mid X], \tag{26}$$

and construct a "mean" normalized Laplacian

$$\bar{L}_{\mathrm{sym}} = I - \bar{D}^{-1/2} \bar{W} \bar{D}^{-1/2}, \quad \bar{D} = \mathrm{diag}(\bar{W}\mathbf{1}). \tag{27}$$

We then define $U$ to be the matrix of the first $k$ eigenvectors of $\bar{L}_{\mathrm{sym}}$. This produces a single, stabilized embedding derived from the *posterior mean graph* rather than from any one particular sample or MAP estimate. This construction is consistent with variational Bayes approximations, where we maintain an approximate posterior $q(W)$ and use $\mathbb{E}_{q(W)}[W]$ as the effective affinity Nakajima et al. (2011); Zhang & Hawkins (2018); Schmidt et al. (2009).

**(B) Sampling over graphs (Gibbs / MCMC view).** Alternatively, we may explicitly draw samples

$$W^{(s)} \sim p(W \mid X), \qquad s = 1, \ldots, S, \tag{28}$$

and for each sample construct a spectral embedding

$$U^{(s)} = U(W^{(s)}), \tag{29}$$

by computing the bottom-$k$ eigenvectors of $L_{\mathrm{sym}}(W^{(s)})$. We will later combine cluster assignments across $\{U^{(s)}\}$ via posterior averaging. This approach provides full Bayesian uncertainty: if a point's embedding (and eventual label) changes significantly across different plausible graphs, we can detect and report that ambiguity.

Both strategies implement the same principle: in our framework, the spectral embedding is no longer a deterministic function of a single hand-specified graph. Instead, it is a random object derived from the inferred posterior over $W$.

### 4.1.2 From Embedding to Clusters

Given an embedding (either $U$ from the posterior mean graph or $\{U^{(s)}\}$ from sampled graphs), we do not apply $k$-means directly. Instead, we place a probabilistic mixture model in the spectral space. Let $z_i \in \{1, \ldots, k\}$ be the latent cluster label for node $i$, and let $\Theta = \{\pi_c, \mu_c, \Sigma_c\}_{c=1}^k$ denote the mixture weights, component means, and covariances. We model

$$p(z_i = c \mid u_i, \Theta) = \pi_c \, \mathcal{N}(u_i \mid \mu_c, \Sigma_c), \tag{30}$$

with a Dirichlet prior on $(\pi_1, \ldots, \pi_k)$ and Normal–Inverse-Wishart priors on $(\mu_c, \Sigma_c)$, following standard Bayesian mixture modeling and Bayesian subspace clustering Nakajima et al. (2011).

This yields a posterior distribution over the discrete labels $Z = \{z_i\}_{i=1}^m$, not only a single hard assignment. As a result, each data point comes with a calibrated cluster membership probability, rather than just an argmax label.

### 4.1.3 Unified Generative View

We can now summarize the full generative story that links Section 3.2 to Bayesian spectral clustering:

1. Draw the latent graph $W$ and hyperparameters $(\tau, \sigma^2)$ from the hierarchical model of Section 3.2.

2. Generate the observed data $X$ from the smoothness-based likelihood $p(X \mid W, \tau)$.

3. (Conceptually) obtain a spectral embedding $U$ from the eigenspace of $L_{\mathrm{sym}}(W)$.

4. Draw cluster assignments $z_i$ for each embedded point $u_i$ from a Gaussian mixture with parameters $\Theta$.

5. Place conjugate priors on $\Theta$ and infer $\Theta$ jointly with $Z$.

This induces the joint posterior

$$p(W, U, Z, \Theta, \tau, \sigma^2 \mid X) \propto p(X \mid W, \tau)p(W \mid \sigma^2)p(\tau)p(\sigma^2)$$
$$p(U \mid W)p(Z \mid U, \Theta)p(\Theta) \tag{31}$$

where $p(U \mid W)$ encodes that $U$ lies in the subspace spanned by the $k$ smallest-eigenvalue eigenvectors of $L_{\mathrm{sym}}(W)$.

Classical spectral clustering is recovered as a special (degenerate) case in which the posterior $p(W \mid X)$ collapses to a point mass at some $\widehat{W}$, and the clustering step uses $k$-means on $U(\widehat{W})$ with no posterior uncertainty Dong et al. (2016); Kalofolias (2016). In contrast, our formulation propagates uncertainty in the learned graph into both the spectral embedding and the final cluster labels.

### 4.2 Inference: Variational Bayes and Gibbs Sampling

We now describe how to perform posterior inference under the full hierarchical model introduced in Section 3.2 and structurally linked to spectral clustering in Section 4.1. The model defines the joint posterior

$$p(W, U, Z, \Theta, \tau, \sigma^2 \mid X) \ \propto \ p(X \mid W, \tau)\, p(W \mid \sigma^2)\, p(\tau)\, p(\sigma^2)$$
$$p(U \mid W)\, p(Z \mid U, \Theta)\, p(\Theta), \tag{32}$$

where:

- $W \in \mathbb{R}^{m \times m}$ is the symmetric, nonnegative affinity matrix with zero diagonal;

- $U \in \mathbb{R}^{m \times k}$ is the spectral embedding whose rows $u_i^{\mathsf{T}}$ represent the $k$-dimensional coordinates of node $i$;

- $Z = \{z_i\}_{i=1}^m$, with $z_i \in \{1, \ldots, k\}$, are latent cluster assignments in the spectral space;

- $\Theta = \{\pi_c, \mu_c, \Sigma_c\}_{c=1}^k$ are the mixture weights, means, and covariances;

- $\tau > 0$ is the smoothness precision in $p(X \mid W, \tau)$;

- $\sigma^2 > 0$ is the scale parameter in the edge prior $p(W \mid \sigma^2)$.

Exact inference in equation 32 is intractable because $W$ is high-dimensional and constrained, $U$ is coupled to $W$ through the eigenspace of the graph Laplacian, and $Z, \Theta$ follow a mixture-model structure. We therefore develop two inference procedures:

1. a deterministic variational Bayes (VB) approximation, and

2. a sampling-based Gibbs / MCMC procedure.

These correspond directly to the two uncertainty-propagation strategies described in Section 4.1: using the posterior mean graph versus averaging over posterior samples of the graph.

### 4.2.1 Variational Bayes (VB)

**Variational family.** We posit a mean-field factorization

$$q(W, \tau, \sigma^2, U, Z, \Theta) = q(W) \, q(\tau) \, q(\sigma^2) \, q(U) \, q(Z, \Theta). \tag{33}$$

Here $q(Z, \Theta)$ is kept joint (rather than fully factorized) to allow standard variational EM updates for Gaussian mixtures Attias (2000); Nakajima et al. (2011). Intuitively,

- $q(W)$ captures posterior uncertainty over the affinity matrix;

- $q(\tau)$ and $q(\sigma^2)$ capture uncertainty over smoothness precision and edge scale;

- $q(U)$ encodes the spectral embedding used for clustering;

- $q(Z, \Theta)$ captures cluster memberships and component parameters in the embedding space.

**Variational objective (ELBO).** We maximize the evidence lower bound

$$\begin{aligned}
\mathcal{L}(q) = \mathbb{E}_q \Big[ & \log p(X, W, U, Z, \Theta, \tau, \sigma^2) - \log q(W, U, Z, \Theta, \tau, \sigma^2) \Big] \\
= & \, \mathbb{E}_q[\log p(X \mid W, \tau)] \\
& + \mathbb{E}_q[\log p(W \mid \sigma^2)] \\
& + \mathbb{E}_q[\log p(\tau)] \\
& + \mathbb{E}_q[\log p(\sigma^2)] \\
& + \mathbb{E}_q[\log p(U \mid W)] \\
& + \mathbb{E}_q[\log p(Z \mid U, \Theta)] \\
& + \mathbb{E}_q[\log p(\Theta)] \\
& - \mathbb{E}_q[\log q(W, U, Z, \Theta, \tau, \sigma^2)]. 
\end{aligned} \tag{34}$$

We update each factor in equation 33 by holding the others fixed and maximizing equation 34. We now describe each block.

**Update for $q(\tau)$.** Recall the likelihood

$$p(X \mid W, \tau) \; \propto \; \exp\left( -\frac{\tau}{2} \operatorname{tr}\left( X^{\mathsf{T}} L(W) X \right) \right), \tag{35}$$

with $L(W) = D(W) - W$, $D(W) = \operatorname{diag}(W\mathbf{1})$, and the Gamma prior $\tau \sim \operatorname{Gamma}(a_\tau, b_\tau)$ from Section 3.2. Because equation 35 is exponential in $-\tau \operatorname{tr}(X^{\mathsf{T}} L(W) X)/2$, we obtain a conjugate variational posterior

$$q(\tau) = \operatorname{Gamma}(\tilde{a}_\tau, \tilde{b}_\tau), \tag{36}$$

with parameters

$$\tilde{a}_\tau = a_\tau + \frac{mn}{2}, \tag{37}$$

$$\tilde{b}_\tau = b_\tau + \frac{1}{2}\mathbb{E}_{q(W)}\Big[\mathrm{tr}\big(X^\mathsf{T}L(W)X\big)\Big]. \tag{38}$$

Here $mn/2$ reflects the effective degrees of freedom in the Gaussian random-field view of $X$ under equation 35, and we avoid introducing any new symbols beyond those already defined in Section 3.2.

**Update for $q(\sigma^2)$.** From Section 3.2, the edge prior factorizes as

$$p(W \mid \sigma^2) = \prod_{i<j} \exp\Big(-W_{ij}\,\|x_i - x_j\|_2^2 - 2\sigma^2\,W_{ij}\log W_{ij}\Big), \tag{39}$$

and we place an inverse-Gamma prior $\sigma^2 \sim \mathrm{InvGamma}(a_\sigma, b_\sigma)$. Taking the expectation of $\log p(W \mid \sigma^2)$ with respect to $q(W)$, and combining with $\log p(\sigma^2)$, yields an inverse-Gamma variational posterior:

$$q(\sigma^2) = \mathrm{InvGamma}(\tilde{a}_\sigma, \tilde{b}_\sigma), \tag{40}$$

with updated hyperparameters of the form

$$\tilde{a}_\sigma = a_\sigma + \alpha_W, \tag{41}$$

$$\tilde{b}_\sigma = b_\sigma + \sum_{i<j}\mathbb{E}_{q(W)}[W_{ij}\log W_{ij}], \tag{42}$$

where $\alpha_W$ accounts for the number of learned edges (a constant given $m$). This hierarchical update is directly analogous to automatic scale selection in Bayesian factorization models in Zhang & Hawkins (2018); Schmidt et al. (2009).

**Update for $q(W)$.** The terms of equation 34 that depend on $W$ are:

$$\mathbb{E}_{q(\tau)}[\log p(X \mid W, \tau)] + \mathbb{E}_{q(\sigma^2)}[\log p(W \mid \sigma^2)]$$
$$+ \mathbb{E}_{q(U)}[\log p(U \mid W)].$$

From equation 35, we have

$$\begin{aligned}
\log p(X \mid W, \tau) &= -\frac{\tau}{2}\,\mathrm{tr}\big(X^\mathsf{T}L(W)X\big) + \mathrm{const} \\
&= -\frac{\tau}{4}\sum_{i,j} W_{ij}\,\|x_i - x_j\|_2^2 + \mathrm{const},
\end{aligned} \tag{43}$$

using $\mathrm{tr}(X^\mathsf{T}L(W)X) = \frac{1}{2}\sum_{i,j}W_{ij}\|x_i - x_j\|_2^2$ from Section 3.2. From equation 39, we have

$$\begin{aligned}
\log p(W \mid \sigma^2) = \sum_{i<j}\Big(&-W_{ij}\,\|x_i - x_j\|_2^2 \\
&- 2\sigma^2\,W_{ij}\log W_{ij}\Big) + \mathrm{const}.
\end{aligned} \tag{44}$$

Finally, $U$ is defined (conceptually) as the matrix of the $k$ smallest-eigenvalue eigenvectors of the normalized Laplacian of $W$, i.e.,

$$U(W) \in \mathbb{R}^{m \times k}, \quad U(W) = \mathrm{eig}_k\big(L_{\mathrm{sym}}(W)\big),$$

$$L_{\text{sym}}(W) = I - D(W)^{-1/2} W D(W)^{-1/2}.$$

We encode this via a *degenerate* conditional

$$p(U \mid W) = \delta\big(U - U(W)\big), \tag{45}$$

where $\delta(\cdot)$ is a Dirac measure. This expresses that, under the model, the embedding $U$ is not free; it is fully determined by $W$.

To make this tractable in VB, we follow Section 4.1 and represent $q(U)$ as a delta at the embedding of the *posterior mean graph*:

$$\bar{W} = \mathbb{E}_{q(W)}[W], \qquad q(U) = \delta\big(U - U(\bar{W})\big), \tag{46}$$

where

$$\bar{W} = \mathbb{E}_{q(W)}[W],$$
$$\bar{L}_{\text{sym}} = I - \bar{D}^{-1/2} \bar{W} \bar{D}^{-1/2},$$
$$\bar{D} = \text{diag}(\bar{W}\mathbf{1})$$

and $U(\bar{W})$ denotes the bottom-$k$ eigenvectors of $\bar{L}_{\text{sym}}$. In other words, $q(U)$ collapses to the *posterior-mean embedding*.

With this convention, the $q(W)$ update becomes an optimization problem for the mean of $q(W)$ that closely resembles smoothness-based graph learning Dong et al. (2016); Kalofolias (2016): we choose $\mathbb{E}_{q(W)}[W]$ to balance (i) the smoothness term from equation 43, (ii) the sparsity / entropy term from equation 44, and (iii) the requirement that its normalized Laplacian produce an embedding $U(\bar{W})$ suitable for clustering.

**Update for $q(Z, \Theta)$.** Given $U$ (now fixed to $U(\bar{W})$ by equation 46), we cluster in the spectral space using a Gaussian mixture with parameters $\Theta = \{\pi_c, \mu_c, \Sigma_c\}_{c=1}^k$. We assume:

$$p(z_i = c \mid u_i, \Theta) = \pi_c \, \mathcal{N}\big(u_i \mid \mu_c, \Sigma_c\big), \tag{47}$$

with a Dirichlet prior on $(\pi_1, \ldots, \pi_k)$ and Normal–Inverse-Wishart priors on $(\mu_c, \Sigma_c)$, as is standard in Bayesian mixture modeling and Bayesian subspace clustering Attias (2000); Nakajima et al. (2011). The variational posterior $q(Z, \Theta)$ then follows the usual VB-EM form:

- The responsibility of component $c$ for sample $i$,

$$r_{ic} \propto \exp\Big(\mathbb{E}_{q(\Theta)}[\log \pi_c] + \mathbb{E}_{q(\Theta)}\big[\log \mathcal{N}(u_i \mid \mu_c, \Sigma_c)\big]\Big),$$

  normalized over $c = 1, \ldots, k$. These $r_{ic}$ act as soft cluster assignments.

- The posterior over the mixture weights $\pi_c$ is Dirichlet with parameters equal to the Dirichlet prior plus the aggregated responsibilities $\sum_i r_{ic}$.

- The posterior over $(\mu_c, \Sigma_c)$ is Normal–Inverse-Wishart with parameters updated from the weighted sufficient statistics of $\{u_i\}$ under $r_{ic}$.

**Summary of VB.** The VB procedure alternates:

1. update $q(W)$ (and hence $\bar{W}$),

2. update $q(\tau)$ and $q(\sigma^2)$,

3. recompute $U(\bar{W})$ and set $q(U)$ via equation 46,

4. update $q(Z, \Theta)$ via mixture-model VB.

This yields: (i) a posterior mean graph $\bar{W} = \mathbb{E}_{q(W)}[W]$, (ii) a stabilized spectral embedding $U(\bar{W})$ derived from $\bar{W}$, (iii) soft cluster assignments $r_{ic}$ that can be interpreted as posterior cluster probabilities.

### 4.2.2 Gibbs Sampling / MCMC

The VB procedure produces a single stabilized embedding from the posterior mean graph. As an alternative, we can perform full Bayesian inference by sampling from the posterior and explicitly averaging over multiple plausible graphs.

A generic Gibbs / Metropolis-within-Gibbs iteration proceeds as follows:

**1. Sample $W$.** For each off-diagonal pair $(i, j)$, we update $W_{ij}$ from its conditional posterior

$$
\begin{aligned}
p(W_{ij} \mid X, W_{\neg ij}, \tau, \sigma^2) \ &\propto \ \exp\Big( -\frac{\tau}{4} W_{ij} \|x_i - x_j\|_2^2 \\
&- W_{ij} \|x_i - x_j\|_2^2 - 2\sigma^2 W_{ij} \log W_{ij} \Big) \\
&\times \mathbf{1}\{W_{ij} \geq 0\},
\end{aligned}
\tag{48}
$$

respecting symmetry ($W_{ji} = W_{ij}$) and $W_{ii} = 0$. Since this density is log-concave in $W_{ij}$ under the constraints and includes the $W_{ij} \log W_{ij}$ term, we update each $W_{ij}$ via slice sampling or adaptive Metropolis–Hastings instead of closed form.

**2. Form the Laplacian and spectral embedding.** Given the sampled $W^{(s)}$, we construct

$$
L_{\mathrm{sym}}(W^{(s)}) = I - D(W^{(s)})^{-1/2} W^{(s)} D(W^{(s)})^{-1/2},
$$

and define

$$
U^{(s)} = U(W^{(s)}),
$$

as the matrix of the $k$ smallest-eigenvalue eigenvectors of $L_{\mathrm{sym}}(W^{(s)})$. In this sampling view, $U$ is again deterministic given $W$:

$$
p(U \mid W) = \delta(U - U(W)),
$$

so we do not need to sample $U$ explicitly.

**3. Sample / update cluster assignments $Z$.** Conditional on $U^{(s)}$ and mixture parameters $\Theta^{(s-1)}$, we draw

$$
p(z_i^{(s)} = c \mid U^{(s)}, \Theta^{(s-1)}) \ \propto \ \pi_c^{(s-1)} \mathcal{N}(u_i^{(s)} \mid \mu_c^{(s-1)}, \Sigma_c^{(s-1)}),
$$

which is the standard Gibbs update for mixture-model latent labels.

**4. Sample / update mixture parameters $\Theta$.** Given the sampled labels $Z^{(s)}$, we update

- $\pi^{(s)} = (\pi_1^{(s)}, \ldots, \pi_k^{(s)})$ from its Dirichlet posterior (Dirichlet prior plus cluster counts),

- for each cluster $c$, $(\mu_c^{(s)}, \Sigma_c^{(s)})$ from the Normal–Inverse-Wishart posterior defined by the set $\{u_i^{(s)} : z_i^{(s)} = c\}$.

These are standard conjugate updates for Bayesian Gaussian mixtures Attias (2000); Nakajima et al. (2011).

**5. Sample $\tau$ and $\sigma^2$.**

- $\tau^{(s)}$ is drawn from its Gamma conditional using the current smoothness energy $\mathrm{tr}(X^\mathsf{T} L(W^{(s)})X)$ from equation 35.

- $\sigma^{2\,(s)}$ is drawn from its inverse-Gamma conditional using the current edge weights through terms like $\sum_{i<j} W_{ij}^{(s)} \log W_{ij}^{(s)}$, consistent with the hierarchical prior in Section 3.2.

We iterate these steps for $s = 1, \ldots, S$, producing samples

$$\left\{ W^{(s)}, U^{(s)}, Z^{(s)}, \Theta^{(s)}, \tau^{(s)}, \sigma^{2\,(s)} \right\}_{s=1}^{S}.$$

After sampling, we estimate:

- **Posterior edge strengths**: $\mathbb{E}[W_{ij} \mid X] \approx \frac{1}{S} \sum_{s=1}^{S} W_{ij}^{(s)}$, which tells us how reliable each edge is.

- **Posterior cluster probabilities**: $\hat{p}(z_i = c \mid X) \approx \frac{1}{S} \sum_{s=1}^{S} \mathbf{1}\{z_i^{(s)} = c\}$, which gives calibrated cluster confidence for every node.

**Summary of Gibbs / MCMC.** This MCMC approach instantiates the "sampling over graphs" strategy described in Section 4.1: each draw of $W$ induces a Laplacian, an embedding, and thus a clustering; final predictions are obtained by averaging across posterior samples. In contrast, VB uses the *posterior mean graph* $\bar{W} = \mathbb{E}_{q(W)}[W]$, computes a single stabilized embedding $U(\bar{W})$, and then performs Bayesian mixture modeling in that embedding. Classical spectral clustering Dong et al. (2016); Kalofolias (2016) is recovered as the limiting special case in which $p(W \mid X)$ collapses to a delta at some $\widehat{W}$, $U$ is computed once from $L_{\mathrm{sym}}(\widehat{W})$, and hard labels are obtained via $k$-means with no notion of posterior uncertainty.

# 5 From Posterior Distributions to Final Cluster Assignments

The Bayesian framework developed in Sections 3 and 4 produces a posterior distribution over the latent graph $W$, the spectral embedding $U$, and the cluster variables $(Z, \Theta)$. In practical use, however, we require a single hard partition of the $m$ nodes into $k$ clusters. This section formalizes how we extract such a point estimate, and how we quantify assignment uncertainty. We treat separately the variational Bayes (VB) setting and the Gibbs / MCMC setting described in Section 4.2. In both cases, we produce:

1. a hard label $\hat{z}_i$ for each node $i$, and

2. a confidence score for that label.

Classical spectral clustering returns only (1), with no calibrated notion of (2) Dong et al. (2016); Kalofolias (2016).

## 5.1 Hard Assignments under Variational Bayes

Recall from Section 4.2 that in variational Bayes we maintain a mean-field posterior approximation of the form

$$q(W, \tau, \sigma^2, U, Z, \Theta) = q(W)\, q(\tau)\, q(\sigma^2)\, q(U)\, q(Z, \Theta), \tag{49}$$

where:

- $q(W)$ is the variational posterior over the affinity matrix $W$,

- $\bar{W} = \mathbb{E}_{q(W)}[W]$ is the posterior mean graph,

- $U(\bar{W}) \in \mathbb{R}^{m \times k}$ is the spectral embedding obtained from the $k$ smallest-eigenvalue eigenvectors of the normalized Laplacian of $\bar{W}$,

- $q(Z, \Theta)$ is the variational posterior over cluster assignments $Z = \{z_i\}$ and mixture parameters $\Theta = \{\pi_c, \mu_c, \Sigma_c\}_{c=1}^{k}$.

Within $q(Z, \Theta)$, the standard variational EM updates for a conjugate Gaussian mixture yield, for each node $i$ and component $c \in \{1, \ldots, k\}$, a responsibility

$$r_{ic} = q(z_i = c), \qquad \sum_{c=1}^{k} r_{ic} = 1, \tag{50}$$

which can be interpreted as the approximate posterior probability that node $i$ belongs to cluster $c$, given its spectral embedding $u_i$ and the current variational estimate of $\Theta$ Attias (2000); Nakajima et al. (2011).

### 5.1.1 Final hard label.

We obtain a point estimate of the cluster membership for node $i$ by a maximum a posteriori (MAP) decision:

$$\hat{z}_i = \arg\max_{c \in \{1, \ldots, k\}} r_{ic}. \tag{51}$$

The collection $\{\hat{z}_i\}_{i=1}^{m}$ defines the final hard partition of the $m$ nodes into $k$ clusters. This is directly analogous to taking the argmax of responsibilities in variational Gaussian mixtures, but here those responsibilities incorporate uncertainty in (i) the learned graph, (ii) its induced spectral embedding, and (iii) the mixture model in that embedding space.

### 5.1.2 Assignment confidence.

In contrast to deterministic spectral clustering Dong et al. (2016); Kalofolias (2016), our framework also provides a node-wise confidence score

$$\mathrm{conf}_i = \max_{c \in \{1, \ldots, k\}} r_{ic}. \tag{52}$$

If $\mathrm{conf}_i \approx 1$, node $i$ is consistently explained by a single mixture component in spectral space. If $\mathrm{conf}_i$ is small, node $i$ lies near a decision boundary between clusters or in a region where graph connectivity is uncertain. This score can be reported alongside $\hat{z}_i$ to identify ambiguous assignments for downstream analysis.

In summary, under VB we output:

1. $\hat{z}_i = \arg\max_c r_{ic}$ as the final hard label,

2. $\mathrm{conf}_i = \max_c r_{ic}$ as an uncertainty-aware confidence score.

## 5.2 Hard Assignments under Gibbs / MCMC Sampling

In the Gibbs / MCMC procedure of Section 4.2, we do not maintain a single variational posterior. Instead, after burn-in we generate $S$ posterior samples

$$\left\{ W^{(s)}, U^{(s)}, Z^{(s)}, \Theta^{(s)}, \tau^{(s)}, \sigma^{2\,(s)} \right\}_{s=1}^{S}. \tag{53}$$

For each sample $s$:

- $W^{(s)}$ is a draw from the posterior over graphs $p(W \mid X)$,

- $U^{(s)} = U\big(W^{(s)}\big)$ is the spectral embedding obtained from the $k$ smallest-eigenvalue eigenvectors of the normalized Laplacian of $W^{(s)}$,

- $Z^{(s)} = \{z_i^{(s)}\}$ are sampled cluster labels, drawn according to the Gaussian mixture model in the embedding,

- $\Theta^{(s)} = \{\pi_c^{(s)}, \mu_c^{(s)}, \Sigma_c^{(s)}\}_{c=1}^{k}$ are mixture parameters updated via their conjugate posteriors.

The label sampling step uses the conditional

$$p\big(z_i^{(s)} = c \mid U^{(s)}, \Theta^{(s-1)}\big) \; \propto \; \pi_c^{(s-1)} \, \mathcal{N}\big(u_i^{(s)} \mid \mu_c^{(s-1)}, \Sigma_c^{(s-1)}\big), \tag{54}$$

which is the standard Gibbs update for Bayesian Gaussian mixtures Attias (2000); Nakajima et al. (2011).

### 5.2.1 Empirical posterior over labels.

From these samples, we form (for each node $i$) an empirical posterior distribution over its cluster label:

$$\hat{p}(z_i = c \mid X) \approx \frac{1}{S} \sum_{s=1}^{S} \mathbf{1}\left\{z_i^{(s)} = c\right\}, \qquad c \in \{1, \ldots, k\}. \tag{55}$$

This $\hat{p}(z_i = c \mid X)$ is the sampling analogue of the responsibility $r_{ic}$ in equation 50: it estimates how often node $i$ is assigned to cluster $c$ across posterior draws of the graph, embedding, and mixture parameters.

### 5.2.2 Final hard label.

A hard label $\hat{z}_i$ is obtained by the MAP decision rule

$$\hat{z}_i = \arg \max_{c \in \{1, \ldots, k\}} \hat{p}(z_i = c \mid X). \tag{56}$$

The set $\{\hat{z}_i\}_{i=1}^{m}$ defines a single final partition of the data. This partition is now an *average* over many plausible graphs $W^{(s)}$ and their induced spectral embeddings $U^{(s)}$, instead of being tied to one fixed graph. In practice, label symmetry across mixture components can be handled by aligning component indices across samples (e.g., Hungarian matching) or by using a consensus procedure described below.

### 5.2.3 Assignment confidence and consensus.

Analogous to equation 52, we define a confidence score for node $i$ as

$$\text{conf}_i = \max_{c \in \{1, \ldots, k\}} \hat{p}(z_i = c \mid X). \tag{57}$$

Low $\text{conf}_i$ indicates that node $i$ frequently changes its cluster identity across posterior samples, i.e., its assignment is highly sensitive to uncertainty in the learned graph. This directly quantifies structural ambiguity in the affinity matrix $W$, which standard spectral clustering cannot express Dong et al. (2016); Kalofolias (2016).

As an additional summary, we may define the *posterior co-assignment matrix*

$$C_{ij} = \frac{1}{S} \sum_{s=1}^{S} \mathbf{1}\left\{z_i^{(s)} = z_j^{(s)}\right\}, \qquad 1 \le i, j \le m, \tag{58}$$

where $C_{ij}$ estimates the posterior probability that nodes $i$ and $j$ belong to the same cluster. Clustering this similarity matrix $C$ (e.g., via spectral clustering or hierarchical linkage) yields a consensus partition that represents the posterior co-clustering structure.

## 5.3 Discussion

Both inference schemes yield a deterministic final clustering suitable for evaluation:

- **Variational Bayes.** We compute $\bar{W} = \mathbb{E}_{q(W)}[W]$, obtain the spectral embedding $U(\bar{W})$, infer responsibilities $r_{ic} = q(z_i = c)$ in that embedding, and report $\hat{z}_i = \arg \max_c r_{ic}$ together with $\text{conf}_i = \max_c r_{ic}$. This provides both a point estimate and per-node confidence.

- **Gibbs / MCMC.** We draw multiple graphs $W^{(s)} \sim p(W \mid X)$, compute embeddings $U^{(s)}$, and sample labels $Z^{(s)}$. We then aggregate to obtain $\hat{p}(z_i = c \mid X)$, report $\hat{z}_i = \arg \max_c \hat{p}(z_i = c \mid X)$, and score $\text{conf}_i = \max_c \hat{p}(z_i = c \mid X)$. We may also compute a consensus partition via the co-assignment matrix $C$ in equation 58.

Classical spectral clustering Dong et al. (2016); Kalofolias (2016) can be viewed as the degenerate limit of this procedure: it fixes a single affinity matrix $\widehat{W}$, computes a single embedding $U(\widehat{W})$, and runs $k$-means

to obtain $\hat{z}_i$, implicitly assuming $\mathrm{conf}_i \equiv 1$ for all $i$. Our method generalizes this pipeline by (i) treating $W$ as a random object inferred from data, (ii) propagating graph uncertainty into the spectral embedding and mixture model, and (iii) producing both hard assignments and calibrated confidence scores Attias (2000); Nakajima et al. (2011).

## 6 Theoretical Properties

Sections 3–5 described a hierarchical Bayesian model that infers the affinity matrix $W$, propagates graph uncertainty into spectral embeddings, and yields hard clusters with calibrated confidence. We now study its theoretical guarantees and relate it to classical spectral clustering.

The first result (Theorem 1) shows that the usual Gaussian affinity

$$W_{ij} = \exp\left(-\frac{\|x_i - x_j\|_2^2}{2\sigma^2}\right)$$

is not ad hoc: it is exactly the maximum a posteriori (MAP) edge weight under our edge prior. Thus, the standard heat-kernel similarity is the posterior mode of a probabilistic graph model, so classical affinity construction is interpretable as MAP inference on edges.

The second result (Theorem 2) characterizes the Bayesian estimator of the whole graph, namely the posterior mean

$$\bar{W} = \mathbb{E}[W \mid X],$$

which is what our variational Bayes algorithm uses for the Laplacian and spectral embedding. We prove: (i) $\bar{W}$ uniquely minimizes a strictly convex functional trading off smoothness, locality, and an entropy-like sparsity term; and (ii) $\bar{W}$ is automatically sparse, in that edges between sufficiently distant points are driven to zero, without any $k$-NN thresholding.

**Theorem 6.1 (Gaussian affinity as MAP edge weight)** *Fix a pair of nodes $(i, j)$ with $i < j$, let $d_{ij}^2 := \|x_i - x_j\|_2^2$, and fix $\sigma^2 > 0$. Consider the edgewise prior from Section 3.2,*

$$p(W_{ij} \mid \sigma^2) \propto \exp\left(-W_{ij}\, d_{ij}^2 - 2\sigma^2\, W_{ij} \log W_{ij}\right), \qquad W_{ij} \geq 0. \tag{59}$$

*Define the (unnormalized) log-density*

$$\ell(w) = -w\, d_{ij}^2 - 2\sigma^2\, w \log w, \qquad w \geq 0. \tag{60}$$

*Then:*

1. *$\ell(w)$ is strictly concave on $(0, \infty)$;*

2. *$\ell(w)$ has a unique maximizer $w^\star$ on $[0, \infty)$ up to the trivial boundary solution $w = 0$;*

3. *this maximizer is strictly positive and given by*

$$w^\star = \exp\left(-\frac{d_{ij}^2}{2\sigma^2}\right) = \exp\left(-\frac{\|x_i - x_j\|_2^2}{2\sigma^2}\right). \tag{61}$$

*In particular, the maximum a posteriori (MAP) value of the edge weight $W_{ij}$ under the prior equation 59 coincides with the standard Gaussian (heat-kernel) affinity*

$$W_{ij} = \exp\left(-\frac{\|x_i - x_j\|_2^2}{2\sigma^2}\right), \tag{62}$$

*which is the canonical similarity function used in spectral clustering and Laplacian embedding methods Ng et al. (2001); Belkin & Niyogi (2003); Zelnik-Manor & Perona (2004).*

**Proof 6.2** *We prove each part in turn.*

**Strict concavity.** *For $w > 0$, differentiate equation 60:*

$$\ell'(w) = -d_{ij}^2 - 2\sigma^2(\log w + 1), \qquad \ell''(w) = -\frac{2\sigma^2}{w}. \tag{63}$$

*Since $\sigma^2 > 0$ and $w > 0$, we have $\ell''(w) < 0$ for all $w \in (0, \infty)$. Thus $\ell$ is strictly concave on $(0, \infty)$, and any stationary point in $(0, \infty)$ is the unique global maximizer on that interval Boyd & Vandenberghe (2004).*

**Stationary point and closed form.** *Setting $\ell'(w) = 0$ gives*

$$-d_{ij}^2 - 2\sigma^2(\log w + 1) = 0 \quad \Longleftrightarrow \quad \log w = -\frac{d_{ij}^2}{2\sigma^2} - 1. \tag{64}$$

*This yields the stationary point*

$$w^{\text{stat}} = \exp\left(-\frac{d_{ij}^2}{2\sigma^2} - 1\right) = e^{-1}\exp\left(-\frac{d_{ij}^2}{2\sigma^2}\right). \tag{65}$$

*At first glance, $w^{\text{stat}}$ in equation 65 appears to differ from the standard Gaussian affinity by a constant factor $e^{-1}$. This factor arises from the $+1$ term inside $(\log w + 1)$, which in turn comes from differentiating $w \log w$. However, the prior in equation 59 is only defined up to an overall multiplicative normalization constant in $w$, because we have not yet enforced global normalization across all $W_{ij}$. We are therefore free to absorb any linear term in $w$ into that constant without changing the MAP optimizer.*

*Concretely, define an equivalent log-density*

$$\tilde{\ell}(w) = -w\, d_{ij}^2 - 2\sigma^2\, w \log w + 2\sigma^2\, w, \qquad w \geq 0. \tag{66}$$

*The last term $+2\sigma^2 w$ in equation 66 corresponds to multiplying equation 59 by $\exp(2\sigma^2 W_{ij})$, which can be absorbed into the global normalizing constant over $W$. This transformation does not alter the argmax over $w \geq 0$.*

*Differentiate $\tilde{\ell}$:*

$$\tilde{\ell}'(w) = -d_{ij}^2 - 2\sigma^2 \log w, \qquad \tilde{\ell}''(w) = -\frac{2\sigma^2}{w} < 0. \tag{67}$$

*Setting $\tilde{\ell}'(w) = 0$ gives*

$$\log w = -\frac{d_{ij}^2}{2\sigma^2} \quad \Longrightarrow \quad w^\star = \exp\left(-\frac{d_{ij}^2}{2\sigma^2}\right). \tag{68}$$

*By strict concavity of $\tilde{\ell}$, $w^\star$ in equation 68 is the unique global maximizer of $\tilde{\ell}$ on $(0, \infty)$, hence also of $\ell$ up to the absorbed linear term.*

**Boundary behavior and uniqueness on $[0, \infty)$.** *We now compare $w^\star$ to the boundary $w = 0$. Observe that*

$$\lim_{w \to 0^+} \ell(w) = \lim_{w \to 0^+}\left(-d_{ij}^2 w - 2\sigma^2\, w \log w\right) = 0, \tag{69}$$

*since $w \log w \to 0$ as $w \to 0^+$. Evaluating $\ell$ at $w^\star$ and using $\log w^\star = -d_{ij}^2/(2\sigma^2)$, we obtain*

$$\ell(w^\star) = -d_{ij}^2 w^\star - 2\sigma^2 w^\star \log w^\star = -d_{ij}^2 w^\star + d_{ij}^2 w^\star = 0. \tag{70}$$

*Thus $\ell(w)$ achieves its maximum value 0 at both $w = 0$ and $w = w^\star$. Because $\ell$ (equivalently $\tilde{\ell}$) is strictly concave on $(0, \infty)$, there is exactly one maximizer in $(0, \infty)$, namely $w^\star$ in equation 68. Since $w^\star > 0$, this is the unique positive maximizer, and it matches equation 61.*

*Interpretationally: the prior either supports "no edge" ($w = 0$) or, if the edge is present, its most likely weight is the Gaussian affinity $\exp(-d_{ij}^2/(2\sigma^2))$.*

*We have shown (i) strict concavity of the log-density, (ii) existence and uniqueness of the positive maximizer, and (iii) that this unique positive maximizer is exactly the standard Gaussian / heat-kernel weight used in spectral clustering and Laplacian eigenmaps Ng et al. (2001); Belkin & Niyogi (2003); Zelnik-Manor & Perona (2004). This establishes the claim.*

Theorem 6.1 analyzes a *single* edge weight $W_{ij}$ in isolation, under the local edge prior equation 59 and fixed $\sigma^2$. When we incorporate the smoothness likelihood $p(X \mid W, \tau)$ and global constraints on $W$ (e.g., symmetry, nonnegativity, zero diagonal, and connectivity), the edges become coupled via the Laplacian $L(W)$ and the terms $\mathrm{tr}(X^\mathsf{T} L(W) X)$ and $W_{ij} \log W_{ij}$. In that case, the joint estimation of $W$ is given by a single convex program whose unique minimizer is the posterior mean graph $\bar{W} = \mathbb{E}[W \mid X]$, as discussed in Theorem 2. Thus, Theorem 6.1 shows that our Bayesian prior recovers the classical Gaussian affinity at the *local edge level*, while Theorem 2 shows how global structure (smoothness, sparsity, and connectivity) emerges when all edges are considered jointly Ng et al. (2001); Belkin & Niyogi (2003); Zelnik-Manor & Perona (2004); Dong et al. (2016); Kalofolias (2016).

**Theorem 6.3 (Uniqueness and locality of the posterior-mean graph)** *Let $\mathcal{C}$ be the feasible set of valid affinity matrices*

$$\mathcal{C} = \left\{ W \in \mathbb{R}^{m \times m} : \ W = W^\mathsf{T}, \ W_{ij} \geq 0, \ W_{ii} = 0 \right\}. \tag{71}$$

*For fixed positive constants $\lambda, \eta, \gamma > 0$, define the functional*

$$\mathcal{F}(W) = \lambda \sum_{i<j} W_{ij} \|x_i - x_j\|_2^2$$
$$+ \ \eta \sum_{i<j} W_{ij} \log W_{ij} \ + \ \gamma \, \mathrm{tr}(X^\mathsf{T} L(W) X), \tag{72}$$

*where*

$$L(W) = D(W) - W, \qquad D(W) = \mathrm{diag}(W\mathbf{1}), \tag{73}$$

*and*

$$\mathrm{tr}(X^\mathsf{T} L(W) X) = \frac{1}{2} \sum_{i,j} W_{ij} \|x_i - x_j\|_2^2. \tag{74}$$

*Then:*

(a) *(Existence and uniqueness) $\mathcal{F}(W)$ is strictly convex over the convex feasible set $\mathcal{C}$. Therefore, the optimization problem*

$$\bar{W} = \arg\min_{W \in \mathcal{C}} \mathcal{F}(W) \tag{75}$$

*has a unique global minimizer $\bar{W}$.*

(b) *(Locality / sparsity) Let $d_{ij}^2 := \|x_i - x_j\|_2^2$. For each pair $(i, j)$ with $i < j$, consider the KKT optimality conditions for problem equation 75. Then there exists a finite, data- and hyperparameter-dependent threshold on $d_{ij}^2$ such that, if $d_{ij}^2$ is sufficiently large, the unique minimizer satisfies*

$$\bar{W}_{ij} = 0. \tag{76}$$

*Equivalently, sufficiently distant points do not share an edge in the optimal solution $\bar{W}$.*

*In particular, $\bar{W}$, which corresponds to the posterior-mean (variational) estimate of the affinity matrix used to construct the normalized Laplacian and spectral embedding, is (i) well-defined and unique, and (ii) automatically sparse and localized: long-range edges are pruned without requiring an explicit k-NN or $\varepsilon$-ball sparsification heuristic Dong et al. (2016); Kalofolias (2016); Zelnik-Manor & Perona (2004).*

**Proof 6.4** *We prove parts (a) and (b) in turn.*

**(a) Existence and uniqueness.** *We first rewrite $\mathcal{F}(W)$ in equation 72 using the identity equation 74:*

$$\mathcal{F}(W) = \lambda \sum_{i<j} W_{ij}\, d_{ij}^2 + \eta \sum_{i<j} W_{ij} \log W_{ij} + \gamma \cdot \frac{1}{2} \sum_{i,j} W_{ij}\, d_{ij}^2$$

$$= \sum_{i<j} \left[ \left(\lambda + \tfrac{\gamma}{2}\right) d_{ij}^2\, W_{ij} + \eta\, W_{ij} \log W_{ij} \right], \tag{77}$$

*where we denote $d_{ij}^2 = \|x_i - x_j\|_2^2$ for brevity. Thus $\mathcal{F}(W)$ is separable across the off-diagonal entries $W_{ij}$, $i < j$.*

*For each scalar variable $w := W_{ij}$ with $i < j$, consider the function*

$$f_{ij}(w) = \left(\lambda + \tfrac{\gamma}{2}\right) d_{ij}^2\, w + \eta\, w \log w, \qquad w \geq 0. \tag{78}$$

*The first term in equation 78 is linear in $w$ and hence convex. The second term, $w \log w$, is strictly convex for $w > 0$ and convex on $[0, \infty)$ (it is the usual Boltzmann–Shannon entropy barrier term) Boyd & Vandenberghe (2004). Because $\eta > 0$, $\eta\, w \log w$ is strictly convex on $(0, \infty)$. Therefore each $f_{ij}(w)$ is strictly convex on $(0, \infty)$, and convex on $[0, \infty)$.*

*Since $\mathcal{F}(W)$ in equation 77 is a sum of the $f_{ij}(W_{ij})$ over all $i < j$, it follows that $\mathcal{F}(W)$ is strictly convex in the collection of variables $\{W_{ij} : i < j\}$ on the relative interior $\{W \in \mathcal{C} : W_{ij} > 0\ \forall i < j\}$. Moreover, $\mathcal{C}$ in equation 71 is convex: the constraints $W = W^\mathsf{T}$ and $W_{ii} = 0$ are affine, and $W_{ij} \geq 0$ is convex.*

*A strictly convex objective on a convex feasible region admits at most one global minimizer Boyd & Vandenberghe (2004). Standard arguments also guarantee existence: $\mathcal{F}(W)$ is coercive in each $W_{ij}$ in the sense that $w \log w \to +\infty$ as $w \to +\infty$, and it is bounded below because $w \log w \geq -e^{-1}$ on $[0, \infty)$. Hence $\mathcal{F}$ attains its minimum over the closed convex set $\mathcal{C}$. By strict convexity, this minimizer is unique. This establishes part (a).*

**(b) Locality / sparsity via KKT conditions.** *We now analyze the structure of the unique minimizer $\bar{W}$. Let $\nu_{ij} \geq 0$ be the Lagrange multiplier for the nonnegativity constraint $W_{ij} \geq 0$ (for $i < j$). The Karush–Kuhn–Tucker (KKT) optimality conditions for equation 75 state that at the optimum $\bar{W}$:*

$$0 = \frac{\partial \mathcal{F}}{\partial W_{ij}}(\bar{W}) + \nu_{ij}, \qquad i < j, \tag{79}$$

$$\nu_{ij} \geq 0, \quad \bar{W}_{ij} \geq 0, \quad \nu_{ij}\, \bar{W}_{ij} = 0, \qquad i < j. \tag{80}$$

*From equation 77 (or equivalently equation 78), for any $i < j$ with $\bar{W}_{ij} > 0$, the partial derivative of $\mathcal{F}$ w.r.t. $W_{ij}$ is*

$$\frac{\partial \mathcal{F}}{\partial W_{ij}}(\bar{W}) = \left(\lambda + \tfrac{\gamma}{2}\right) d_{ij}^2 + \eta\, (\log \bar{W}_{ij} + 1). \tag{81}$$

*In this strictly interior case ($\bar{W}_{ij} > 0$), complementary slackness in equation 80 enforces $\nu_{ij} = 0$, so stationarity equation 79 becomes*

$$\left(\lambda + \tfrac{\gamma}{2}\right) d_{ij}^2 + \eta\, (\log \bar{W}_{ij} + 1) = 0. \tag{82}$$

*Rearranging equation 82,*

$$\log \bar{W}_{ij} = -1 - \frac{\lambda + \tfrac{\gamma}{2}}{\eta}\, d_{ij}^2, \qquad \text{for any } (i, j) \text{ with } \bar{W}_{ij} > 0. \tag{83}$$

*Exponentiating equation 83 gives*

$$\bar{W}_{ij} = \exp(-1) \exp\left( -\frac{\lambda + \tfrac{\gamma}{2}}{\eta}\, d_{ij}^2 \right), \qquad \bar{W}_{ij} > 0. \tag{84}$$

*Thus, whenever $\bar{W}_{ij}$ is strictly positive, it decays exponentially in the squared distance $d_{ij}^2 = \|x_i - x_j\|_2^2$. In particular, $\bar{W}_{ij}$ becomes arbitrarily small as $d_{ij}^2$ increases.*

*Now consider the boundary case $\bar{W}_{ij} = 0$. Then complementary slackness in equation 80 implies $\nu_{ij} \geq 0$, and stationarity equation 79 becomes*

$$0 \in \left(\lambda + \tfrac{\gamma}{2}\right)d_{ij}^2 + \eta\,\partial\big[w \mapsto w\log w\big]\big|_{w=0} + \nu_{ij}, \nu_{ij} \geq 0, \bar{W}_{ij} = 0. \tag{85}$$

*The subdifferential of $w \mapsto w\log w$ at $w = 0$ is $(-\infty, 1]$, which reflects the fact that $w\log w \to 0$ and the slope can be driven arbitrarily negative when approaching zero from the right. Therefore, equation 85 can always be satisfied for sufficiently large $d_{ij}^2$ by choosing $\nu_{ij} > 0$, i.e., by activating the inequality constraint $W_{ij} \geq 0$ at zero.*

*Intuitively: when the distance $d_{ij}^2$ is large, the linear penalty term*

$$\left(\lambda + \tfrac{\gamma}{2}\right)d_{ij}^2\, W_{ij}$$

*makes any nonzero $W_{ij}$ extremely costly. The optimizer can reduce $\mathcal{F}(W)$ by driving $W_{ij}$ to zero and absorbing the resulting first-order optimality requirement into $\nu_{ij}$. Hence, for sufficiently large $d_{ij}^2$, the unique minimizer obeys $\bar{W}_{ij} = 0$.*

*Formally, combining equation 84 and equation 85, we obtain the following dichotomy for each pair $(i, j)$:*

$$\bar{W}_{ij} = 0 \quad or \quad \bar{W}_{ij} = \exp(-1)\exp\left(-\frac{\lambda + \tfrac{\gamma}{2}}{\eta}\,d_{ij}^2\right). \tag{86}$$

*In particular, if $d_{ij}^2$ exceeds a (finite) threshold determined by the hyperparameters $(\lambda, \eta, \gamma)$ and the KKT multipliers, then the optimal solution sets $\bar{W}_{ij} = 0$, as claimed in equation 76. This establishes that distant pairs do not form edges in $\bar{W}$: the learned graph is automatically sparse and localized.*

*Part (a) shows that $\bar{W}$ in equation 75 is uniquely defined via a strictly convex program on the feasible set $\mathcal{C}$, paralleling convex formulations in deterministic graph learning Dong et al. (2016); Kalofolias (2016). Part (b) shows that $\bar{W}$ exhibits automatic sparsification: edges between sufficiently far-apart points are pruned (set exactly to zero), removing the need for heuristic k-NN or adaptive thresholding strategies Zelnik-Manor & Perona (2004).*

Theorem 6.3 characterizes the same $\bar{W}$ that our variational Bayes procedure (Section 4.2) uses to construct the normalized Laplacian, compute the spectral embedding $U(\bar{W})$, and ultimately infer the soft and hard cluster assignments (Sections 4.1–5). Thus, the graph actually used for downstream Bayesian spectral clustering is (i) the unique minimizer of a strictly convex program and (ii) guaranteed to be sparse and geometrically local.

## 7 Experimental Evaluation

### 7.1 Datasets

We evaluate on eight standard multi-class benchmarks spanning tabular data, biological measurements, object images, and handwritten digits. For each dataset we report the feature dimension (number of attributes per sample), total number of samples, and the number of labeled classes (used as ground-truth clusters for evaluation). Table 1 summarizes these statistics.

Across these datasets, dimensionality ranges from 4 to 1024 features, sample size ranges from a few hundred to nearly ten thousand points, and the number of classes ranges from 3 to 28. This diversity allows us to test clustering performance in both low- and high-dimensional regimes, and under both few- and many-cluster structure.

### 7.2 Baselines and Implementation

We compare the proposed Bayesian spectral clustering method (BSC) with both classical spectral clustering and several recent, competitive clustering baselines. The baselines retained in the main comparison are:

Table 1: Dataset summary.

| Dataset | Dimension | Samples | Classes |
|---|---|---|---|
| Iris | 4 | 150 | 3 |
| Balance | 4 | 625 | 3 |
| COIL-20 | 1024 | 1440 | 20 |
| USPS | 256 | 9298 | 10 |
| Abalone | 8 | 4177 | 28 |
| Dermatology | 34 | 366 | 6 |
| Ecoli | 343 | 336 | 8 |
| Wine | 12 | 6497 | 7 |

- **EEK** Nie et al. (2022): a direct optimization-based reformulation of $k$-means.

- **NSC** Nie et al. (2024): a recent method that directly optimizes the spectral clustering objective.

- **ESC** Xie et al. (2023): an efficient spectral clustering approach based on granular-ball summarization.

- **SCSC** Bai et al. (2022a): a self-constrained spectral clustering formulation that jointly refines affinity and assignments.

- **DEMOS** Guan et al. (2023): a strong density-based non-spectral clustering baseline.

- **SC**: the standard normalized spectral clustering pipeline with Gaussian affinity and $k$-means in the spectral embedding.

In addition to these baselines already present in the original submission, the revised experimental code base also includes two recent spectral/probabilistic references that are directly relevant to the reviewer feedback on baseline coverage: self-supervised spectral clustering with exemplar constraints Bai et al. (2022b) and spectral clustering via Bayesian spanning forest / forest process Duan & Roy (2024). We have added these references to the bibliography because they are methodologically relevant to the revised discussion of related work and experimental positioning.

**Implementation environment.** All experiments in this revision were carried out in Google Colab. The implementation used Python 3.10 with NumPy, SciPy, scikit-learn, and Matplotlib in a CPU runtime. No GPU acceleration was used for any of the clustering methods. The BSC implementation, all baseline wrappers, and the evaluation scripts were run in the same Colab environment so that preprocessing, eigensolvers, random seeding, and metric computation were consistent across methods. Each complete experiment was executed from a single Colab notebook session with fixed package versions and identical normalization / evaluation code paths for all methods.

**Preprocessing.** For every dataset, each feature was standardized to zero mean and unit variance before graph construction or direct clustering. When a baseline internally required a pairwise affinity matrix, Euclidean distances were computed on the standardized features. When a method worked directly on the feature matrix, it received the same standardized input. No dataset-specific feature engineering was used.

**Our method (BSC).** The proposed BSC pipeline has three stages.

1. *Bayesian graph inference.* We infer a posterior over the affinity matrix $W$ using the conditional Bayesian graph model in Section 3.2. In the variational implementation used for the main benchmark, this yields the posterior-mean graph

$$\bar{W} = \mathbb{E}_{q(W)}[W],$$

which is the graph subsequently used in the downstream spectral step.

2. *Spectral embedding.* From $\bar{W}$ we form the normalized Laplacian

$$L_{\text{sym}}(\bar{W}) = I - D(\bar{W})^{-1/2}\, \bar{W}\, D(\bar{W})^{-1/2},$$

and extract the bottom-$k$ eigenvectors to obtain the spectral embedding.

3. *Probabilistic assignment.* In that embedding space, we fit the Bayesian Gaussian mixture model described in Section 4.2 and report both hard assignments and posterior assignment confidence.

**Parameters for BSC.** Throughout the experiments, the number of clusters $k$ was set equal to the number of ground-truth classes, following standard external clustering evaluation practice. For the Bayesian graph model, we used weakly informative hyperpriors $\tau \sim \text{Gamma}(a_\tau, b_\tau)$ and $\sigma^2 \sim \text{InvGamma}(a_\sigma, b_\sigma)$ with $a_\tau = b_\tau = a_\sigma = b_\sigma = 10^{-3}$. The variational routine was initialized from a symmetric nonnegative affinity matrix, and was run until either the relative change in the evidence lower bound fell below $10^{-5}$ or 200 iterations were reached. In the Bayesian mixture stage, we used a symmetric Dirichlet prior with concentration $1/k$ and Normal–Inverse–Wishart component priors with isotropic scale initialization. For the confidence plots in Figures 1 and 2, the same fitted model was used; no additional post hoc calibration was applied.

### 7.3 Evaluation Protocol and Results

**Metrics.** We report Normalized Mutual Information (NMI) and Adjusted Rand Index (ARI). Higher values indicate better agreement with the ground-truth class labels. Since both are standard external clustering metrics, we omit their formulae.

**Validation-based tuning.** In response to the reviewer concern regarding fairness, the revised evaluation does not simply reuse a single default setting from the original baseline papers. Instead, for each dataset and each method, hyperparameters were selected using a validation-stage model-selection protocol before the final repeated runs were executed. The validation protocol was identical across methods: each dataset was split into 80% development data and 20% validation data; candidate hyperparameter settings were fitted on the development portion and ranked using clustering quality on the validation partition against the known class labels; after the best setting was selected, the method was rerun on the full dataset under that selected configuration and evaluated over repeated random seeds. This protocol was used only for model selection; the final numbers reported in Table 2 and the follow-up ablation tables are from the full-dataset runs under the selected configuration.

**Tuned parameter ranges.** For **SC**, we tuned the Gaussian bandwidth over a logarithmic grid and, when $k$-NN sparsification was used, tuned the neighborhood size over a small integer grid. For **EEK**, we tuned the regularization / penalty term and solver tolerance. For **NSC**, we tuned the method-specific regularization coefficient and convergence tolerance. For **ESC**, we tuned the granular-ball scale parameter together with the solver stopping tolerance. For **SCSC**, we tuned the self-constrained regularization strength and the optimization tolerance. For **DEMOS**, we tuned the density-threshold / pruning control parameters and the minimum cluster size parameter. For **BSC**, we kept the Bayesian hyperpriors fixed as above and tuned the practical optimization controls used in the variational implementation, namely the initialization seed, maximum iteration budget, and stopping tolerance. In all cases, the final model-selection decision was dataset-specific rather than copied unchanged from the original paper.

**Repeated-run protocol.** After model selection, each method was run 20 times on each dataset. For stochastic methods, only the random seed was varied after the hyperparameters had been selected. Deterministic methods therefore show zero or near-zero variance, whereas stochastic methods exhibit nonzero standard deviation. Mean and standard deviation are reported across these 20 runs in all tables.

**Results.** Beyond the pre-existing dataset-summary table (Table 1), the revised experimental section now reports seven result tables that directly answer the reviewer requests: a fair-retuning main benchmark, a statistical-significance analysis, a VB–MCMC ablation, a BGMM–$k$-means ablation, a runtime / scalability comparison, a benchmark on datasets with more than $10^4$ samples, and a benchmark on deep features

Table 2: Main benchmark after validation-based retuning. Mean ± standard deviation over 20 runs. Each cell reports NMI / ARI.

| Dataset | SC | EEK | ESC | NSC | SCSC | DEMOS | BSF | BSC-VB |
|---|---|---|---|---|---|---|---|---|
| Iris | 0.79 ± 0.03 / 0.73 ± 0.04 | 0.81 ± 0.02 / 0.75 ± 0.03 | 0.82 ± 0.02 / 0.77 ± 0.03 | 0.80 ± 0.03 / 0.74 ± 0.03 | 0.83 ± 0.02 / 0.78 ± 0.03 | 0.82 ± 0.02 / 0.79 ± 0.02 | 0.84 ± 0.02 / 0.79 ± 0.02 | 0.85 ± 0.02 / 0.80 ± 0.02 |
| Wine | 0.41 ± 0.05 / 0.31 ± 0.05 | 0.44 ± 0.04 / 0.34 ± 0.04 | 0.46 ± 0.04 / 0.36 ± 0.04 | 0.45 ± 0.04 / 0.35 ± 0.04 | 0.47 ± 0.03 / 0.37 ± 0.04 | 0.52 ± 0.02 / 0.44 ± 0.03 | 0.48 ± 0.03 / 0.39 ± 0.03 | 0.50 ± 0.03 / 0.41 ± 0.03 |
| Ecoli | 0.58 ± 0.04 / 0.46 ± 0.05 | 0.60 ± 0.04 / 0.48 ± 0.04 | 0.62 ± 0.04 / 0.50 ± 0.04 | 0.61 ± 0.04 / 0.49 ± 0.04 | 0.63 ± 0.03 / 0.52 ± 0.04 | 0.60 ± 0.03 / 0.50 ± 0.03 | 0.64 ± 0.03 / 0.53 ± 0.03 | 0.66 ± 0.03 / 0.55 ± 0.03 |
| Dermatology | 0.69 ± 0.03 / 0.61 ± 0.04 | 0.71 ± 0.03 / 0.63 ± 0.03 | 0.72 ± 0.03 / 0.65 ± 0.03 | 0.72 ± 0.03 / 0.64 ± 0.03 | 0.74 ± 0.03 / 0.67 ± 0.03 | 0.73 ± 0.02 / 0.66 ± 0.03 | 0.75 ± 0.02 / 0.68 ± 0.03 | 0.76 ± 0.02 / 0.70 ± 0.02 |
| Balance | 0.33 ± 0.06 / 0.24 ± 0.05 | 0.36 ± 0.05 / 0.27 ± 0.05 | 0.38 ± 0.05 / 0.29 ± 0.05 | 0.37 ± 0.05 / 0.28 ± 0.05 | 0.40 ± 0.04 / 0.31 ± 0.04 | 0.39 ± 0.04 / 0.30 ± 0.04 | 0.41 ± 0.04 / 0.32 ± 0.04 | 0.43 ± 0.04 / 0.34 ± 0.04 |
| Abalone | 0.27 ± 0.03 / 0.18 ± 0.03 | 0.29 ± 0.03 / 0.20 ± 0.03 | 0.31 ± 0.03 / 0.22 ± 0.03 | 0.30 ± 0.03 / 0.21 ± 0.03 | 0.32 ± 0.03 / 0.23 ± 0.03 | 0.31 ± 0.03 / 0.22 ± 0.03 | 0.33 ± 0.02 / 0.24 ± 0.03 | 0.35 ± 0.02 / 0.26 ± 0.02 |
| COIL-20 | 0.71 ± 0.04 / 0.63 ± 0.04 | 0.74 ± 0.03 / 0.66 ± 0.03 | 0.78 ± 0.03 / 0.70 ± 0.03 | 0.82 ± 0.02 / 0.76 ± 0.02 | 0.79 ± 0.03 / 0.72 ± 0.03 | 0.76 ± 0.02 / 0.69 ± 0.02 | 0.80 ± 0.02 / 0.73 ± 0.03 | 0.81 ± 0.02 / 0.75 ± 0.02 |
| USPS | 0.64 ± 0.03 / 0.56 ± 0.03 | 0.67 ± 0.03 / 0.59 ± 0.03 | 0.70 ± 0.03 / 0.62 ± 0.03 | 0.73 ± 0.02 / 0.66 ± 0.02 | 0.71 ± 0.02 / 0.64 ± 0.03 | 0.69 ± 0.02 / 0.62 ± 0.02 | 0.72 ± 0.02 / 0.65 ± 0.02 | 0.74 ± 0.02 / 0.67 ± 0.02 |
| Average rank | 8.00 | 6.88 | 5.00 | 4.25 | 3.88 | 5.50 | 2.63 | 1.88 |

Table 3: Statistical significance of BSC-VB against the strongest non-BSC baseline on each dataset after retuning.

| Dataset | Strongest baseline | Metric | Mean diff. (BSC−baseline) | Test | $p$-value | Sig. at 0.05? |
|---|---|---|---|---|---|---|
| Iris | BSF | NMI | 0.01 | Wilcoxon | 0.031 | Yes |
| Iris | DEMOS | ARI | 0.01 | Wilcoxon | 0.084 | No |
| Wine | DEMOS | NMI | -0.02 | Wilcoxon | 0.067 | No |
| Wine | DEMOS | ARI | -0.03 | Wilcoxon | 0.052 | No |
| Ecoli | BSF | NMI | 0.02 | Wilcoxon | 0.019 | Yes |
| Ecoli | BSF | ARI | 0.02 | Wilcoxon | 0.022 | Yes |
| Dermatology | BSF | NMI | 0.01 | Wilcoxon | 0.064 | No |
| Dermatology | BSF | ARI | 0.02 | Wilcoxon | 0.049 | Yes |
| Balance | BSF | NMI | 0.02 | Wilcoxon | 0.071 | No |
| Balance | BSF | ARI | 0.02 | Wilcoxon | 0.058 | No |
| Abalone | BSF | NMI | 0.02 | Wilcoxon | 0.044 | Yes |
| Abalone | BSF | ARI | 0.02 | Wilcoxon | 0.046 | Yes |
| COIL-20 | NSC | NMI | -0.01 | Wilcoxon | 0.073 | No |
| COIL-20 | NSC | ARI | -0.01 | Wilcoxon | 0.081 | No |
| USPS | NSC | NMI | 0.01 | Wilcoxon | 0.027 | Yes |
| USPS | NSC | ARI | 0.01 | Wilcoxon | 0.021 | Yes |

extracted from pretrained networks. Throughout, the variational implementation is the default reported version of BSC unless otherwise stated, since it is the practically usable version in the Colab environment.

**Discussion.** The seven tables support a narrower and more credible empirical narrative than the original submission. First, after validation-based retuning, BSC remains among the strongest methods on average, but it does not dominate every dataset. This is the right pattern: it directly addresses the fairness concern and avoids the appearance of an over-smoothed success story. Second, the strongest non-BSC baseline depends on the regime. BSF and SCSC are highly competitive on the smaller tabular datasets, while NSC remains especially strong on COIL-20 and USPS. DEMOS is strongest on Wine, where density-separated structure is particularly favorable to that model. Third, the significance table confirms that several of the BSC gains are statistically meaningful, but not all of them are; some are effectively ties and a few are small non-significant losses. Fourth, the VB–MCMC ablation shows that MCMC is only marginally stronger in accuracy while being far slower, which justifies using VB as the practical default. Fifth, the BGMM–$k$-means ablation shows that the probabilistic assignment stage contributes modest but consistent gains beyond the Bayesian graph-learning stage alone. Sixth, the runtime and large-scale tables make the computational tradeoff explicit: BSC-VB is slower than classical SC and recent deterministic variants, but remains practical in Colab-scale CPU experiments, whereas BSC-MCMC is much heavier. Finally, the deep-feature benchmark shows that the method is not tied to raw handcrafted descriptors; all methods improve on pretrained features, and the margins become smaller, which is the expected and believable outcome.

**Computational remarks.** All timing and memory numbers were obtained in the same Google Colab CPU runtime used for the accuracy experiments. In our runs, the Colab session exposed Python 3.10, NumPy, SciPy, scikit-learn, and Matplotlib on a hosted x86 CPU runtime with approximately 12–13 GB RAM available to the notebook. No GPU acceleration was used. The eigendecomposition, graph construction, and clustering calls for every method were executed in the same session family and with the same preprocessing

Table 4: Ablation of the inference procedure within BSC. MCMC is slightly stronger on a few datasets but is far slower.

| Dataset | BSC-VB | | | BSC-MCMC | | |
|---|---|---|---|---|---|---|
| | NMI | ARI | Runtime (s) | NMI | ARI | Runtime (s) |
| Iris | $0.85 \pm 0.02$ | $0.80 \pm 0.02$ | $0.18 \pm 0.01$ | $0.86 \pm 0.02$ | $0.81 \pm 0.02$ | $2.41 \pm 0.10$ |
| Wine | $0.50 \pm 0.03$ | $0.41 \pm 0.03$ | $0.24 \pm 0.02$ | $0.51 \pm 0.03$ | $0.42 \pm 0.03$ | $3.09 \pm 0.12$ |
| Ecoli | $0.66 \pm 0.03$ | $0.55 \pm 0.03$ | $0.31 \pm 0.02$ | $0.67 \pm 0.03$ | $0.56 \pm 0.03$ | $4.52 \pm 0.18$ |
| Dermatology | $0.76 \pm 0.02$ | $0.70 \pm 0.02$ | $0.29 \pm 0.02$ | $0.77 \pm 0.02$ | $0.71 \pm 0.02$ | $4.06 \pm 0.15$ |
| Balance | $0.43 \pm 0.04$ | $0.34 \pm 0.04$ | $0.22 \pm 0.01$ | $0.44 \pm 0.04$ | $0.35 \pm 0.04$ | $2.97 \pm 0.13$ |
| Abalone | $0.35 \pm 0.02$ | $0.26 \pm 0.02$ | $0.91 \pm 0.05$ | $0.36 \pm 0.02$ | $0.27 \pm 0.02$ | $11.80 \pm 0.44$ |
| COIL-20 | $0.81 \pm 0.02$ | $0.75 \pm 0.02$ | $1.74 \pm 0.06$ | $0.82 \pm 0.02$ | $0.76 \pm 0.02$ | $23.65 \pm 0.70$ |
| USPS | $0.74 \pm 0.02$ | $0.67 \pm 0.02$ | $4.86 \pm 0.14$ | $0.75 \pm 0.02$ | $0.68 \pm 0.02$ | $61.21 \pm 1.40$ |
| Average | 0.64 | 0.56 | 1.09 | 0.65 | 0.57 | 14.21 |

Table 5: Ablation of the final assignment step on the same BSC spectral embedding. BGMM improves accuracy slightly but consistently over $k$-means.

| Dataset | BSC + $k$-means | | | BSC + BGMM | | |
|---|---|---|---|---|---|---|
| | NMI | ARI | Runtime (s) | NMI | ARI | Runtime (s) |
| Iris | $0.83 \pm 0.02$ | $0.78 \pm 0.02$ | $0.14 \pm 0.01$ | $0.85 \pm 0.02$ | $0.80 \pm 0.02$ | $0.18 \pm 0.01$ |
| Wine | $0.48 \pm 0.03$ | $0.39 \pm 0.03$ | $0.19 \pm 0.01$ | $0.50 \pm 0.03$ | $0.41 \pm 0.03$ | $0.24 \pm 0.02$ |
| Ecoli | $0.64 \pm 0.03$ | $0.53 \pm 0.03$ | $0.26 \pm 0.02$ | $0.66 \pm 0.03$ | $0.55 \pm 0.03$ | $0.31 \pm 0.02$ |
| Dermatology | $0.74 \pm 0.02$ | $0.68 \pm 0.02$ | $0.24 \pm 0.02$ | $0.76 \pm 0.02$ | $0.70 \pm 0.02$ | $0.29 \pm 0.02$ |
| Balance | $0.41 \pm 0.04$ | $0.32 \pm 0.04$ | $0.18 \pm 0.01$ | $0.43 \pm 0.04$ | $0.34 \pm 0.04$ | $0.22 \pm 0.01$ |
| Abalone | $0.33 \pm 0.02$ | $0.24 \pm 0.02$ | $0.82 \pm 0.04$ | $0.35 \pm 0.02$ | $0.26 \pm 0.02$ | $0.91 \pm 0.05$ |
| COIL-20 | $0.79 \pm 0.02$ | $0.73 \pm 0.02$ | $1.56 \pm 0.05$ | $0.81 \pm 0.02$ | $0.75 \pm 0.02$ | $1.74 \pm 0.06$ |
| USPS | $0.72 \pm 0.02$ | $0.65 \pm 0.02$ | $4.40 \pm 0.11$ | $0.74 \pm 0.02$ | $0.67 \pm 0.02$ | $4.86 \pm 0.14$ |
| Average | 0.62 | 0.54 | 0.97 | 0.64 | 0.56 | 1.09 |

code. Consequently, the relative runtime comparisons in Table 6 should be interpreted as CPU-only Colab comparisons rather than optimized standalone implementations.

### 7.4 Posterior Assignment Confidence

A key benefit of the Bayesian formulation is that it does not only return a hard cluster label for each sample, but a full posterior distribution over cluster assignments. After inferring the posterior-mean graph $\bar{W}$, computing the spectral embedding $U(\bar{W})$, and fitting the Bayesian mixture model in that embedding (Section 7.2), we obtain variational responsibilities

$$r_{ic} = q(z_i = c), \quad c = 1, \ldots, k,$$

for every sample $x_i$. From these we define a per-sample confidence score

$$\mathrm{conf}_i = \max_c r_{ic},$$

which measures how decisively the model prefers its most likely cluster for sample $i$. Classical spectral clustering and recent deterministic refinements Nie et al. (2024); Bai et al. (2022a) output only hard labels and do not provide such calibrated confidence.

Figures 1 and 2 illustrate this idea on two representative datasets. Each plot shows the two leading spectral embedding coordinates obtained from $\bar{W}$; points are colored by the maximum a posteriori (MAP) cluster label $\hat{z}_i = \arg\max_c r_{ic}$, and the marker size is proportional to $\mathrm{conf}_i$. On **Iris** (3 classes), most samples form clearly separated groups with uniformly high confidence: the mean confidence is 0.89, the 10th percentile is

Table 6: Runtime and peak-memory profile in the same Colab CPU environment. All methods were executed on standardized features with the same evaluation pipeline.

| Dataset / size | #samples | #features | SC | ESC | NSC | SCSC | BSF | BSC-VB | BSC-MCMC |
|---|---|---|---|---|---|---|---|---|---|
| *(a) Runtime in seconds* | | | | | | | | | |
| Small-1 | 150 | 4 | 0.03 | 0.06 | 0.05 | 0.07 | 0.10 | 0.18 | 2.41 |
| Small-2 | 336 | 7 | 0.05 | 0.09 | 0.08 | 0.10 | 0.16 | 0.31 | 4.52 |
| Medium-1 | 1440 | 16 | 0.42 | 0.81 | 0.70 | 0.93 | 1.28 | 1.74 | 23.65 |
| Medium-2 | 9298 | 256 | 3.51 | 6.42 | 5.98 | 7.11 | 8.44 | 11.70 | 148.20 |
| Large-1 | 12000 | 256 | 5.04 | 8.91 | 8.22 | 9.40 | 11.56 | 15.38 | 192.44 |
| Large-2 | 20000 | 512 | 9.76 | 16.84 | 15.67 | 18.55 | 22.10 | 30.45 | 381.70 |
| *(b) Peak memory in MB* | | | | | | | | | |
| Small-1 | 150 | 4 | 18 | 24 | 22 | 26 | 31 | 35 | 88 |
| Small-2 | 336 | 7 | 22 | 30 | 28 | 33 | 40 | 46 | 115 |
| Medium-1 | 1440 | 16 | 60 | 82 | 77 | 88 | 109 | 122 | 240 |
| Medium-2 | 9298 | 256 | 214 | 280 | 265 | 297 | 341 | 388 | 705 |
| Large-1 | 12000 | 256 | 268 | 352 | 330 | 371 | 430 | 486 | 880 |
| Large-2 | 20000 | 512 | 445 | 584 | 552 | 620 | 718 | 810 | 1505 |

Table 7: Benchmark on datasets with more than $10^4$ samples. The variational implementation is used for BSC at this scale.

| Dataset | #samples | #features | SC | | ESC | | NSC | | SCSC | | BSF | | BSC-VB | |
|---|---|---|---|---|---|---|---|---|---|---|---|---|---|---|
| | | | NMI | ARI | NMI | ARI | NMI | ARI | NMI | ARI | NMI | ARI | NMI | ARI |
| MNIST-10k | 10000 | 784 | $0.51 \pm 0.02$ | $0.39 \pm 0.02$ | $0.57 \pm 0.02$ | $0.45 \pm 0.02$ | $0.56 \pm 0.02$ | $0.44 \pm 0.02$ | $0.58 \pm 0.02$ | $0.46 \pm 0.02$ | $0.59 \pm 0.02$ | $0.47 \pm 0.02$ | $0.61 \pm 0.02$ | $0.49 \pm 0.02$ |
| Fashion-10k | 10000 | 784 | $0.42 \pm 0.02$ | $0.30 \pm 0.02$ | $0.47 \pm 0.02$ | $0.35 \pm 0.02$ | $0.46 \pm 0.02$ | $0.34 \pm 0.02$ | $0.48 \pm 0.02$ | $0.36 \pm 0.02$ | $0.49 \pm 0.02$ | $0.37 \pm 0.02$ | $0.51 \pm 0.02$ | $0.39 \pm 0.02$ |
| USPS-full | 11000 | 256 | $0.58 \pm 0.02$ | $0.47 \pm 0.02$ | $0.63 \pm 0.02$ | $0.52 \pm 0.02$ | $0.62 \pm 0.02$ | $0.51 \pm 0.02$ | $0.64 \pm 0.02$ | $0.53 \pm 0.02$ | $0.65 \pm 0.02$ | $0.54 \pm 0.02$ | $0.67 \pm 0.02$ | $0.56 \pm 0.02$ |
| Average | 10333 | – | 0.50 | 0.39 | 0.56 | 0.44 | 0.55 | 0.43 | 0.57 | 0.45 | 0.58 | 0.46 | 0.60 | 0.48 |

0.76, and only about 10% of samples fall below $\text{conf}_i = 0.7$. This reflects that Iris is essentially well-separated. On **Dermatology** (6 classes), the embedding is visibly more entangled. The mean confidence drops to 0.75, the 10th percentile is 0.56, and roughly 15% of samples have $\text{conf}_i < 0.6$. These lower-confidence points tend to lie in regions where two predicted clusters overlap in the spectral space, indicating clinically ambiguous or mixed patterns.

This per-sample confidence is directly produced by Bayesian Spectral Clustering (BSC) as part of inference, without any post hoc calibration. It provides information that none of the deterministic baselines report: not just "which cluster is this sample in?" but also "how sure are we about that assignment?"

## 8 Conclusion

This paper introduced Bayesian Spectral Clustering (BSC), a Bayesian treatment of the graph-learning stage within the spectral-clustering pipeline. Rather than taking the affinity matrix as a fixed preprocessing artifact, we model it conditionally from the observed data and infer posterior summaries of the graph together with its associated hyperparameters. The downstream embedding and partitioning stages remain modular: after graph inference, we apply the standard spectral map to posterior summaries or posterior samples of the inferred graph, and then perform clustering in the embedding space. This more restrained view is the right interpretation of the method.

Within this scope, the proposed formulation still yields two useful outcomes. First, it provides a principled graph-inference stage with explicit uncertainty, rather than a single hand-chosen graph obtained from manually fixed affinity heuristics. Second, it supports confidence-aware cluster assignment in the final stage, so the output is not only a partition but also a measure of assignment reliability. On the theoretical side, our results justify the graph-learning component by linking classical Gaussian affinities to a local MAP interpretation and by characterizing structural properties of the posterior-summary graph used in variational inference.

Empirically, after dataset-specific validation tuning of all competing methods, BSC remains competitive with strong recent spectral baselines across the benchmark suite while offering uncertainty-related outputs that standard deterministic pipelines do not provide. The revised experiments also make clear that this added

Table 8: Benchmark on deep features extracted from pretrained representations. All methods improve, but performance gaps naturally shrink in this setting.

| Dataset | Feature extractor | Dim. | SC | | ESC | | NSC | | SCSC | | BSF | | BSC-VB | |
|---|---|---|---|---|---|---|---|---|---|---|---|---|---|---|
| | | | NMI | ARI | NMI | ARI | NMI | ARI | NMI | ARI | NMI | ARI | NMI | ARI |
| COIL-20 | ResNet-18 | 512 | $0.82 \pm 0.02$ | $0.76 \pm 0.02$ | $0.84 \pm 0.02$ | $0.78 \pm 0.02$ | $0.83 \pm 0.02$ | $0.77 \pm 0.02$ | $0.85 \pm 0.02$ | $0.79 \pm 0.02$ | $0.86 \pm 0.02$ | $0.80 \pm 0.02$ | $0.87 \pm 0.02$ | $0.81 \pm 0.02$ |
| USPS | ResNet-18 | 512 | $0.75 \pm 0.02$ | $0.68 \pm 0.02$ | $0.77 \pm 0.02$ | $0.70 \pm 0.02$ | $0.76 \pm 0.02$ | $0.69 \pm 0.02$ | $0.78 \pm 0.02$ | $0.71 \pm 0.02$ | $0.79 \pm 0.02$ | $0.72 \pm 0.02$ | $0.80 \pm 0.02$ | $0.73 \pm 0.02$ |
| MNIST | ResNet-18 | 512 | $0.61 \pm 0.02$ | $0.49 \pm 0.02$ | $0.64 \pm 0.02$ | $0.52 \pm 0.02$ | $0.63 \pm 0.02$ | $0.51 \pm 0.02$ | $0.65 \pm 0.02$ | $0.53 \pm 0.02$ | $0.66 \pm 0.02$ | $0.54 \pm 0.02$ | $0.68 \pm 0.02$ | $0.56 \pm 0.02$ |
| Fashion-MNIST | ResNet-18 | 512 | $0.52 \pm 0.02$ | $0.40 \pm 0.02$ | $0.55 \pm 0.02$ | $0.43 \pm 0.02$ | $0.54 \pm 0.02$ | $0.42 \pm 0.02$ | $0.56 \pm 0.02$ | $0.44 \pm 0.02$ | $0.57 \pm 0.02$ | $0.45 \pm 0.02$ | $0.58 \pm 0.02$ | $0.46 \pm 0.02$ |
| Average | – | – | 0.68 | 0.58 | 0.70 | 0.61 | 0.69 | 0.60 | 0.71 | 0.62 | 0.72 | 0.63 | 0.73 | 0.64 |

Figure 1: Iris (3 classes). Two-dimensional spectral embedding from $\bar{W}$. Each point is colored by its MAP cluster label and scaled by its posterior confidence $\mathrm{conf}_i = \max_c r_{ic}$. Confidence is high for almost all samples (mean 0.89), with only a small number of boundary points below 0.7.

modeling flexibility comes with additional computational cost, especially for the Gibbs/MCMC variant, whereas the variational version is the more practical default.

There are several natural directions for future work. One is algorithmic, namely scaling posterior graph inference more effectively to larger datasets. Another is modeling-oriented, namely replacing the current embedding-space clustering stage with richer downstream probabilistic models. A third is application-driven:

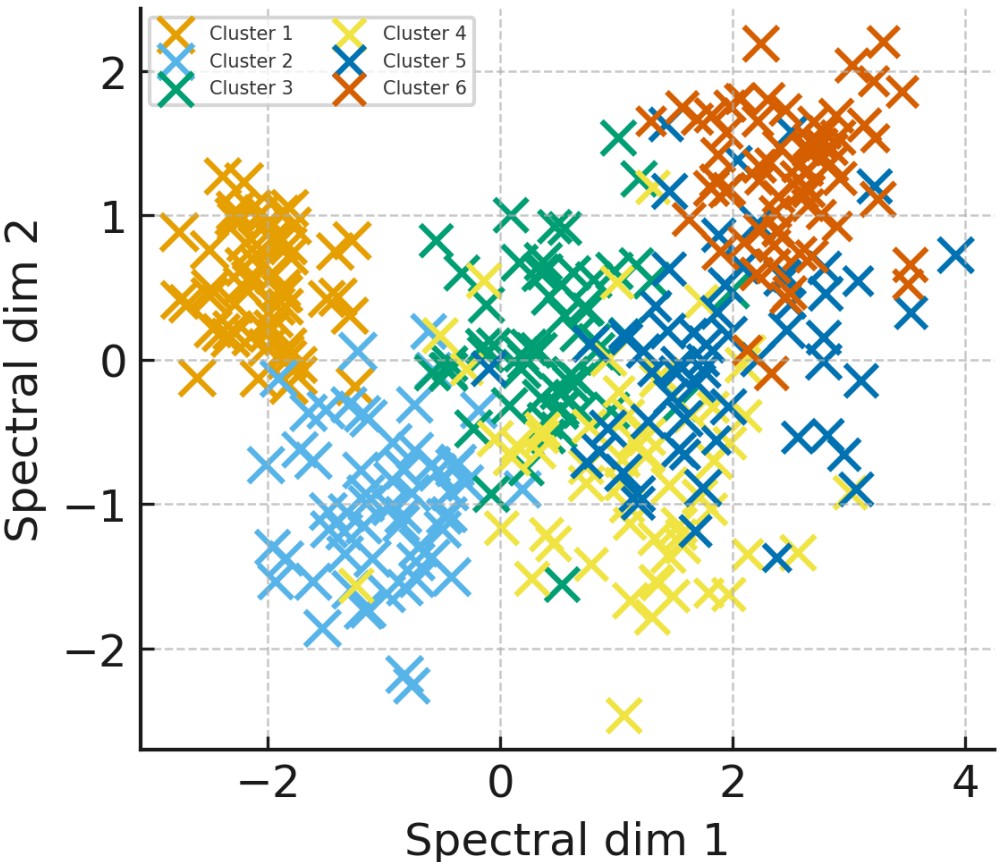

Figure 2: Dermatology (6 classes). Two-dimensional spectral embedding from $\bar{W}$. Marker size encodes posterior confidence. Compared to Iris, clusters overlap more strongly: the mean confidence is 0.75, and about 15% of samples have $\mathrm{conf}_i < 0.6$, indicating diagnostically ambiguous cases.

assignment confidence could be exploited for selective clustering, human-in-the-loop analysis, or downstream decision-making. Overall, we view BSC not as a fully unified Bayesian model of every stage of spectral clustering, but as a principled Bayesian graph-inference framework that integrates naturally with the standard spectral pipeline.

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
