# OpenReview forum: "Bayesian Spectral Clustering"
_TMLR — Rejected by TMLR_

### Review · Reviewer_3z44 · 2026-03-17

**Summary Of Contributions:**

The paper proposes a Bayesian approach to spectral clustering (BSC) based on variational inference and MCMC sampling as two procedures for inference. The approach is based on imposing a prior on the similarity matrix for which its posterior can be sampled (using slice sampling) or represented in terms of expectation (VB). A point estimate of the embedding is subsequently derived and a GMM used to probabilistically assign observation to clusters as opposed to the conventional k-means step of spectral clustering. Theoretical properties of the weight matrix posterior (Theorem 6.1) and uniqueness and sparsity (Theorem 6.3) is derived. In the experimentation favorable performance is found for the proposed BSC when compared to establishes existing spectral clustering procedure in terms of normalized mutual information to class labels (NMI) and adjusted rand index (ARI).

**Additional Comments:**

Given the above paper needs to be substantially revised to be suitable for publication.

**Audience:**

Yes

**Audience Explanation:**

Spectral clustering is a widely used technique and the Bayesian approach here taken appears useful. However, the presentation and organization can be substantially improved (see requested changes).

**Broader Impact Concerns:**

There are no immediate broader impact concerns.

**Claims And Evidence:**

No

**Claims Explanation:**

The manuscript needs to include systematic ablations to understand the impact of the different components introduced (see requested changes)

**Requested Changes:**

Key strengths:
The approach appears useful and accounting for the uncertainty of parameters in spectral clustering including the affinity graph is interesting.
The approach appears well performant when compared to existing spectral clustering procedures (statistical assessments here missing).

Central weaknesses:
The presentation of the methodology and the organization of the manuscript can be substantially improved.
The approach appears to have a substantial computational overhead – especially the MCMC inference procedure that is the most principled.
Many necessary ablations are completely missing from the manuscript.

Detailed comments:

Organization
The paper is in need of a substantial reorganization as it is overly and unnecessary long (see also comments below).

Remove repetitions, i.e. the uncertainty of the affinity matrix contribution is discussed many times throughout the manuscript but only needs to be mentioned once in section 3.

The generative process in section 4.1.3 and presentation in section 3.2.2 is flawed as I see it and this needs to be addressed. In particular, equation 13 is dependent on X inducing a distribution p(W|X,\sigma^2). As a result equation 20 is also incorrect and so is step 1 of the generative process in section 4.1.3 due to the X dependence. This needs to be addressed before the model can be considered meaningful from a generative perspective in my view.

Section 3.2.2 already establish the MAP estimate of W_ij. Theorem 6.1 should be moved to here and the proof given in a supplementary (the proof is also rather straightforward and just lengthy in the main paper).

Numbering is also off. Theorem 6.1 should have Proof 6.2, Theorem 6.3 be Theorem 6.2. with proof 6.2. These proofs I also suggest be moved to section 3.2.2 to complete the discussion of properties of W_ij in one place of the manuscript.

First two terms can trivially be combined in equation 13 using equation 12. For clarity I suggest this be shown explicitly.
q(U) is literally based on a point estimate. It is unclear to me why q(U) can not be inferred probabilistically using tools from the literature such as:
Duan, Leo L., George Michailidis, and Mingzhou Ding. "Bayesian spiked Laplacian graphs." Journal of Machine Learning Research 24.3 (2023): 1-35.
At least this should be further discussed.

Some equations are taking overly much space and are trivial, such as equation 34. The whole discussion of how to achieve hard assignments is also taking much space while being rather trivial. Consider placing this in the supplementary.

Figure 1 and Figure 2 should be combined into a single figure. The figures really do not convey much in my opinion as all markers appear to have same size thus the confidence is not possible to observe rendering the figures without much information to convey.

Overall, the paper can easily be reformatted to fit much fewer main pages leaving some details to the supplementary and tightening up the presentation substantially.

Experimentation:
The manuscript needs to include careful ablations of the different contributions of the BSC methodology. This includes
i)	A comparative analysis of the VB inference procedure to the MCMC procedure.
ii)	Ablations of different model components, i.e. GMM vs. standard K-means for the assignment step to see if it is the uncertainty of W that improves upon results or the probabilistic clustering step by GMM.
iii)	Scalability discussions and experimentations. The methodology includes some additional computational overheads. While these appear to be negligible for the VB the more principled MCMC procedures introduces a sampling overhead of an additional invoked cost of the S samples. This needs to be carefully discussed and experimentally assessed.
iv)    Performance needs to be statistically assessed in terms of significance, currently only error bars are included without statistical assessments.

The choice of methods based on IEEE TKDE and IEEE TPAMI appears arbitrary. Rather cite the methods used regardless of venue for their publications. I would also remove the focus of IEEE TKDE end of section 2.3 and 2.4. The specific journal is not what is important. Same goes for bullet 3 on page 5 and end of page 6 where TKDE is explicitly again mentioned as a venue.

---

> ### Author Response · Authors · 2026-03-25
> **Reviewer 3z44**
>
> Thank you for pointing out the broader issues of overstatement and repetition. We revised the manuscript to reduce both. In particular, we no longer present the pipeline as a fully unified generative Bayesian treatment of graph inference, embedding, and final assignment. Instead, the paper now adopts a more careful modular reading: posterior inference is performed over the graph, the spectral map is applied to posterior summaries or samples, and clustering is then carried out in the embedding space.
> We also took the safer route on the generative-story issue. Rather than insisting on a fully generative interpretation for every stage, the revised manuscript presents the key construction more cautiously and avoids overstating what is being modeled probabilistically. At the same time, we retained the overall paper organization, since we believe presenting the method first and then collecting the structural guarantees in a dedicated theory section is still defensible. We did, however, tighten the wording so that the scope of the theoretical claims is now much clearer.
> The section on converting posterior quantities into final assignments was shortened substantially. It now focuses on the core operational rules for producing hard labels and confidence scores, rather than extended discussion that was not necessary in the main paper.
> The experimental section was also revised to be more systematic. The manuscript now describes a validation-stage tuning procedure followed by final repeated runs, and it expands the evaluation suite to include seven result tables: the retuned main benchmark, statistical significance testing, VB-vs-MCMC, BGMM-vs-k-means, runtime and memory scalability, large-scale data, and deep-feature benchmarks. We also added recent relevant baselines to the bibliography and rewrote the results discussion so that it explicitly allows for ties and dataset-dependent strengths instead of presenting an overly smooth success narrative.
> The conclusion was shortened and brought into line with this narrower scope, ending with a more precise statement of contribution, limitations, and future directions.

---

### Review · Reviewer_JARc · 2026-03-19

**Summary Of Contributions:**

This paper studies spectral clustering, and the main focus is to give a new formulation from a Bayesian perspective. The input X \in R^{m \times n}  is a collection of m points each associated with n features. Roughly, in their formulation, the similarity matrix W, which is usually taken as Gaussian kernel (of the input points) as in standard spectral clustering, is interpreted as a random object, and the formulation gives how to evaluate the posterior Pr[W | X], where X is the input dataset. The final clustering is solved via minimizing a Bayesian loss.

Experiments are conducted on several real-world datasets and compared with various baselines. The result seems to show that the new approach is comparable to existing ones.

Overall, the formulation is new, and the modeling seems to the valid. The experiment results suggest that this is a viable method.

**Audience:**

Yes

**Audience Explanation:**

This paper is about spectral clustering, which is a fundamental problem, and there is rich literature on this topic. This paper adds to that, and from a new Bayesian perspective, which is likely to be interesting to the community.

**Broader Impact Concerns:**

None.

**Claims And Evidence:**

No

**Claims Explanation:**

- The experiment is not convincing. From the experiment results, it seems that the new approach is only comparable to existing ones, not outperforming.

- The analysis of computational complexity is missing. In fact, it would also be good to see this in the experiments. In my opinion, one of the reasons that the standard formulation is used so widely, is that it is easy to implement and runs fast. Hence, for the new formulation, this efficiency is a key point to discuss.

- There are several details in the modeling not well explained, and is not convincing. For instance, how do we handle the new hyper-parameters \sigma and \tau? In particular, this is roughly discussed in Sec 3.2.3, but it is not explained why do we put Gamma (distribution)? Also, the exact technical definition of Gamma and InvGamma are not provided.

- I personally find your high-level motivation for considering Bayesian spectral clustering confusing. You mentioned that you wish to introduce dynamics/probabilistic view, rather than finding a fixed solution, to spectral clustering, and pose this as a "drawback" of the classic method. I think this is not a drawback of classic method; instead, you just take an independent/incomparable approach. For example, the classic spectral clustering may be thought of in a PAC learning sense: the dataset is a sample from a distribution, and finding the "deterministic" clustering on this sample is a natural way for learning the true clustering. Your Bayesian perspective just takes a different approach/framework, and is incomparable.

- Lastly, somewhere in the paper mentions that the new approach does not need the parameter k, which is correct, but it seems new parameters are introduced, and it is unclear if this eventually overcomes the issue of parameter finding.

**Requested Changes:**

I suggest to address my points listed in the explanation why the claim is not clear/convincing.

---

> ### Author Response · Authors · 2026-03-25
> **Reviewer JARc**
>
> Thank you for the comments on framing and parameterization. We revised the paper so that it no longer treats determinism itself as the flaw in classical spectral clustering. The motivation is now narrower: the main issue is that graph construction is often heuristic and uncertainty in the graph is typically not represented, even though it affects the downstream embedding and partition. We also clarify the parameter story more carefully by distinguishing graph-construction heuristics from quantities that remain user-specified, such as the number of clusters.
> We also tightened the modelling discussion. In the revised manuscript, the Bayesian treatment is described more carefully in terms of graph inference and its associated hyperparameters, rather than as a universal removal of all modelling choices. The theory section is correspondingly framed as justifying the graph-learning component of the method, not a full joint posterior treatment of every downstream step.
> Your request for computational clarity led to substantial revision of the experimental section. We now state explicitly that all experiments were run in Google Colab using Python 3.10 with NumPy, SciPy, scikit-learn, and Matplotlib in a CPU runtime, with identical preprocessing and evaluation code paths across methods. We also separate the practical variational implementation from the slower Gibbs/MCMC reference implementation, and we moderate the empirical wording so that the paper now claims competitiveness after fair tuning rather than uniform dominance.
> The conclusion was revised in the same spirit. It now emphasizes uncertainty-aware graph inference within the spectral-clustering pipeline and summarizes the empirical findings in more measured terms.

---

### Review · Reviewer_7tzT · 2026-03-20

**Summary Of Contributions:**

This paper presents a Bayesian formulation of spectral clustering. Instead of computing a single affinity matrix W, typically with a Gaussian kernel and k-nearest neighbor sparsification, it assumes it as a latent random variable with priors that encourage locality and sparsity. Then, variational inference on W and the latent clusters is applied to find a posterior distribution.
This leads to: i) the typically used Gaussian kernel is obtained as maximum a posteriori on the graph edges; ii) the mean-posterior is obtained as the unique solution of a minimization of a convex objective, and it is sparse, without the need for the k-nn sparsification. The final distribution over clusters is computed as a Bayesian Gaussian Mixture model. The presented results seem to be competitive with recent SOTA methods.

Strengths:
- The method is well motivated with clear examples of the generality of the approach, and basic examples of clustering approaches that are generalized by the proposed approach.
- By estimating a posterior distribution over W, the proposed approach can provide more stable information about the graph edges, and therefore a notion of certainty about the estimated edges.
- Sparsity, instead of being a heuristic factor, is a consequence of the proposed formulation when performing variational inference.

Weaknesses:
- Going from the affinity matrix to the spectral embedding is a pointwise deterministic operation based on estimating the eigenvectors of the normalized Laplacian. This breaks the Bayesian inference, and the follow-up Bayesian Gaussian Mixture model is not really connected with the previous part. Thus, in my undestanding the proposed method is not really a single probabilistic approach as claimed, but still the combination of a Bayesian approach on the estimation of the affinity matrix and a separate Bayesian mixture model.
- The evaluation is performed with the hyperparameter setting defined in the original paper. In my opinion, this is not a fair evaluation as hyperparameters should be optimized for the specific data.

**Audience:**

Yes

**Audience Explanation:**

The contributions of this paper might be interesting for multiple TMLR audiences. The proposed approach adds robustness on the estimation of W and generalises previous approaches. In addition, it justifies the use of Gaussian kernels for measuring pair-wise distances as a MAP of the estimated posterior. Finally, the competitive results (assuming that they also hold when optimising the hyperparameters for each method) can make the method useful for practical applications.

**Broader Impact Concerns:**

No ethical implications

**Claims And Evidence:**

Yes

**Claims Explanation:**

Most of the claims of the submission are supported by clear evidence.
The only claim that I think is not supported by evidence is that the method is a single Bayesian approach to spectral clustering. Instead, I see it as a combination of two independent Bayesian approaches, one on the estimation of W and a second one on the Bayesian mixture clustering. The authors should comment on that.

**Requested Changes:**

- (critical) The authors should either better explain why they consider the proposed method as a single end-to-end Bayesian method (see above) or remove the claims about that
- (crititcal) Experimental results should be performed with the hyper-paramaters of each method optimised on a validation set.
- (non critical) References often repeat the author names twice. This should be corrected.
- (non critical) It would be intersting to evaluate the performance of the proposed approach on lerger dataset with more than 10k samples.
- (non critical) It would be interesting to test the clustering algorithm also on features extracted from a Deep Learning method.

---

> ### Author Response · Authors · 2026-03-25
> **Reviewer 7tzT**
>
> Thank you for highlighting that the manuscript was overclaiming the degree of Bayesian unification. We revised the paper to adopt a safer modular interpretation throughout. The Bayesian component is now described as posterior inference over the affinity graph and its hyperparameters, while the spectral embedding is obtained from posterior summaries or samples of that graph and the final clustering is performed downstream in the embedding space. We made this clarification in the methodological discussion, the theory framing, and the conclusion so that the paper no longer reads as claiming one fully end-to-end Bayesian latent-variable model over all stages.
> We also revised the experimental section to address your fairness concern directly. In the revised manuscript, baselines are no longer described as being run only with paper-recommended settings; instead, each method is tuned using a validation-stage model-selection protocol before final reporting. We now state the main tuned parameter classes explicitly, and we frame the empirical outcome more carefully as competitive performance after fair retuning rather than blanket superiority.
> Finally, the revised experimental section is now much more explicit about setup and reporting. It states that the experiments were run in Google Colab in a common CPU-only Python environment, and it introduces the expanded evaluation suite, including the retuned main benchmark, statistical testing, VB-vs-MCMC and BGMM-vs-k-means ablations, scalability analysis, a large-scale benchmark, and a deep-feature benchmark.

---

### Decision · Action_Editor_sTiY · 2026-05-13

**Recommendation:** Reject

**Additional Comments:**

While I commend the authors for their substantial empirical updates during the rebuttal phase—which successfully addressed the concerns of Reviewers 7tzT and JARC regarding experimental baselines and computational overhead—I must ultimately base my decision on the unresolved technical flaws highlighted by Reviewer 3z44. In their response, the authors claimed to adopt a "safer modular interpretation" to resolve the fundamental circularity of their generative process. However, the revised manuscript still retains the mathematically flawed joint posterior equation that circularly depends on the data $X$. Adding textual caveats about a "pipeline" while leaving the underlying problematic formulation intact is insufficient for a rigorous probabilistic framework. Furthermore, the authors declined reasonable requests to improve the manuscript's readability and flow, such as moving dense proofs to the supplementary material. Because the core mathematical claims regarding the generative process remain structurally flawed and uncorrected, the manuscript fails to meet TMLR's criteria for claims being supported by accurate and clear evidence, resulting in a decision to reject.

**Audience:**

Yes

**Audience Explanation:**

Developing a probabilistic, Bayesian reformulation of spectral clustering is a highly relevant problem for the TMLR community. While the paper's mathematical formulation ultimately fell short of acceptance criteria, attempts to quantify uncertainty and dynamically learn affinity graphs are of significant interest to researchers in unsupervised learning and probabilistic modeling. Additionally, the empirical findings—such as the scalability limits and trade-offs between Variational Bayes and MCMC—provide valuable insights that others working on similar graphical models would find useful.

**Claims And Evidence:**

No

**Claims Explanation:**

I answered "no" to this criterion due to unresolved concerns with the paper's mathematical formulation. As highlighted during the review process, the proposed generative process contains a circular dependence on the data $X$. While the authors updated the text to describe the framework as a "modular pipeline," the joint posterior equation in the manuscript was left unchanged and still reflects this circularity. Because the mathematical formulation does not fully align with the paper's theoretical claims, I feel the evidence provided is not sufficiently accurate and clear to meet TMLR's standards.

**Resubmission Of Major Revision:**

The authors may consider submitting a major revision at a later time.